
# Application of Gauss's Theorem to quantify localized surface emissions from airborne measurements of wind and trace gases

Stephen Conley*[1,6], Ian Faloona[1], Shobhit Mehrotra[1], Maxime Suard[1], Donald H. Lenschow[2], Colm Sweeney[4], Scott Herndon[3], Stefan Schwietzke[4,5], Gabrielle Pétron[4,5], Justin Pifer[6], Eric A. Kort[7] and Russell Schnell[5]

[1]Department of Land, Air, & Water Resources, University of California, Davis, 95616, USA
[2]Mesoscale and Microscale Meteorology Laboratory, National Center for Atmospheric Research, Boulder, Colorado, 80307, USA
[3]Aerodyne Research, Inc, Billerica, Massachusetts, 01821, USA
[4]Cooperative Institute for Research in Environmental Sciences, University of Colorado, Boulder, Colorado, 80305, USA
[5]NOAA Earth System Research Laboratory, Boulder, Colorado, USA
[6]Scientific Aviation, Inc., Boulder, CO, USA
[7]Climate and Space Sciences and Engineering, University of Michigan

Correspondence to: S. A. Conley (saconley@ucdavis.edu)



**Abstract**
Airborne estimates of greenhouse gas emissions are becoming more prevalent with the advent of rapid
commercial development of trace gas instrumentation featuring increased measurement accuracy, precision,
and frequency, and the swelling interest in the verification of current emission inventories. Multiple airborne
studies have indicated that emission inventories may underestimate some hydrocarbon emission sources in
U.S. oil and gas producing basins.  Consequently, a proper assessment of the accuracy of these airborne
methods is crucial to interpreting the meaning of such discrepancies. We present a new method of sampling
surface sources of any trace gas for which fast and precise measurements can be made and apply it to
methane, ethane, and carbon dioxide on spatial scales of ~1000 m, where consecutive loops are flown around
a targeted source region at multiple altitudes. Using Reynolds decomposition for the scalar concentrations,
along with Gauss's Theorem, we show that the method accurately accounts for the smaller scale turbulent
dispersion of the local plume, which is often ignored in other average "mass balance" methods. With the help
of large eddy simulations (LES) we further show how the circling radius can be optimized for the
micrometeorological conditions encountered during any flight. Furthermore, by sampling controlled releases
of methane and ethane on the ground we can ascertain that the accuracy of the method, in appropriate
meteorological conditions, is better than 20%, with limits of detection below 5 kg hr$^{-1}$ for both methane and
ethane. Because of the FAA mandated minimum flight safe altitude of 150 m, placement of the aircraft is
critical to preventing a large portion of the emission plume from flowing underneath the lowest aircraft
sampling altitude, which is generally the leading source of uncertainty in these measurements. Finally, we
show how the accuracy of the method is strongly dependent on the number of sampling loops, or time spent
sampling the source plume.
**1 Introduction**
Accurate national inventories of greenhouse gas emissions (primarily carbon dioxide ($CO_2$), methane ($CH_4$), and
nitrous oxide ($N_2O$)) is of paramount importance in developing strategies to understand global emissions. The
multitude of sources, however, are so often highly variable in physical size, emission magnitude, height above
ground, and duration that rigorous verification is exceedingly difficult. Nevertheless, measurement techniques
have improved markedly in the past decade, and these are being employed to an unprecedented extent in an
effort to evaluate and refine emission inventories (Nisbet and Weiss, 2010). Most so called "bottom-up"
inventories are developed by aggregating statistical correlates of individual process emissions to such mapping
variables as population density, energy consumption, head of cattle, etc., extrapolating to total emissions using
a relatively small number of direct measurements. On the other hand, atmospheric scientists have long striven
to use measurements from global surface networks, aircraft campaigns, and satellites to try to determine
emissions based on the amounts and build-up rates of observed trace gases. The latter, "top-down" approach
conveniently integrates the multitudes of sources, but is heavily reliant on a detailed knowledge of
atmospheric transport. Top-down methods also suffer from difficulties attributing sources and generalizing
measurements made over a relatively short time period.  Attempts to reconcile these two distinct methods on
global (Muhle et al., 2010) and continental scales (Gerbig et al., 2003; Miller et al., 2013) have often indicated
an apparent underestimation by the "bottom-up" methods of a factor 1.5 or more.



In principle, the aircraft top-down measurements can be conducted at all of the atmospheric scales to better
understand and identify the emissions at comparable scales. For long-lived greenhouse gases, which readily
disperse throughout the atmosphere, the global scale is very instructive. The seminal experiment began with
Keeling's acclaimed $CO_2$ curve [1960], and has continued through more contemporary techniques by (Hirsch et
al., 2006; Neef et al., 2010) for $CH_4$ and $N_2O$, respectively. At progressively smaller scales more details of the
source strengths and apportionment can be made: from synoptic or continental scales which can help
constrain national inventories (Bergamaschi et al., 2005) or specific biogeographic regions (Gallagher et al.,
1994), to mesoscale investigations to estimate emissions from urban areas (Mays et al., 2009; Turnbull et al.,
2011; Wecht et al., 2014) or specific oil and gas producing fields (Karion et al., 2013; Petron et al., 2014) and
even down to individual point/area sources on the order of 10-100 m size (Denmead et al., 1998; Lavoie et al.,
2015; Roscioli et al., 2015).
Aircraft in-situ measurements are particularly useful for "top-down" methods at the subregional scales
because they can be used to measure the air both upwind and downwind of a source region. However,
deployments tend to be costly and thus sporadic. The aircraft methods used so far tend to be one of three
main types. First, there is the eddy covariance technique that is carried out at low altitudes wherein the
vertical fluxes of gases carried by the turbulent wind are measured by tracking rapid fluctuations of both
concentrations and vertical wind  (Hiller et al., 2014; Ritter et al., 1994; Yuan et al., 2015). This method is
generally thought to be the most direct, but it is limited to small footprint regions which must be repeatedly
sampled for sufficient statistical confidence, requires a sophisticated vertical wind measurement, and can be
subject to errors due to flux divergence between the surface and the lowest flight altitude or sensitivity of the
gas sensor to aircraft acceleration. The second, and by far the most common approach is what chemists usually
refer to as "mass balance" and what is known in the turbulence community as a "scalar budget" technique.
Many different sets of assumptions and sampling strategies are employed, but overall an attempt is made to
sample the main dispersion routes of the surface emissions as they make their way into the overlying
atmosphere after first accumulating in the boundary layer. The scales that can be addressed by this method
are from a few kilometers (Alfieri and Blanken, 2012; Hacker et al., 2016; Hiller et al., 2014; Tratt et al., 2014)
to tens of kilometers (Caulton et al., 2014; Karion et al., 2013; Wratt et al., 2001) to even potentially hundreds
of kilometers (Beswick et al., 1998; Chang et al., 2014), and this approach has been the focus of recent
measurements in natural gas production basins. These basins present a source apportionment challenge in
that emissions from multiple sources (agriculture, oil & gas wells, geologic seepage, etc.) commingle as the air
mass travels across the basin. The third general method of source quantification is by referencing
measurements in the atmosphere to another trace gas with a metered release (tracer) or otherwise known
emission rate and assuming that the tracer and the scalar of interest have the same diffusion characteristics.
Typically this tracer release technique is applied to small scales of tens to hundreds of meters (Czepiel et al.,
1996; Lamb et al., 1995; Roscioli et al., 2015), but the principle has been attempted at the basin (Peischl et al.,
2013) and continent (Miller et al., 2012) scales using a reference trace gas with suitable known emission rates
such as $CO_2$ or CO.
The airborne mass balance flight strategies can be grouped into three basic patterns: a single height transect
around a source assuming a vertically uniformly mixed boundary layer (Karion et al., 2013); single height
upwind/downwind (Wratt et al., 2001) or sometimes just downwind flight legs (Conley et al., 2016; Hacker et
al., 2016; Ryerson et al., 1998); and multiple flight legs at different altitudes, either in a stacked box
configuration (Alfieri et al., 2012; Gordon et al., 2015; Kalthoff et al., 2002); or just a 'screen' on the downwind
face of the box (Karion et al., 2015; Lavoie et al., 2015; Mays et al., 2009).



Here we describe a new airborne method borne out of a necessity to identify and quantify source emissions to within 20% accuracy in a large heterogeneous field of potential sources. The novel technique applies an aircraft flight pattern that circumscribes a virtual cylinder around an emission source and, using only observed horizontal wind and trace gas concentrations, applies Gauss's Theorem to estimate the flux divergence through that cylinder. By integrating the outward horizontal fluxes at each point along the circular flight path, the flux contributions from upwind sources can be accounted for. Making an accurate estimate, however, requires the selection of an appropriate circling radius based on the micrometeorological conditions inferred in flight from measurements onboard the aircraft. The pattern must be far enough downstream for the plume to mix sufficiently in the vertical, yet not so far that the trace gas plume enhancements do not stand out sufficiently from the background concentration.

In this study we present the general analytical method used to derive emission estimates using airborne measurements. After the underlying theory is outlined, we investigate the structure of a dispersing plume using large-eddy simulation (LES) to better understand the optimal sampling strategies for quantifying near surface gas sources. We then evaluate the accuracy of the approach using coordinated planned release experiments and by applying the method to $CO_2$ from several power plant plumes to compare with reported emissions.

## 2 Data Collection

### 2.1 Airborne Instrumentation

The airborne detection system is flown on a fixed wing single engine Mooney aircraft, extensively modified for research as described in (Conley et al., 2014). Ambient air is collected through ~5 m of tubing (Kynar, Teflon and stainless steel) that protrudes out of backward-facing aluminum inlets mounted below the right wing. In-situ $CH_4$, $CO_2$, and water vapor are measured with a Picarro 2301f cavity ring down spectrometer as described by (Crosson, 2008), which is operated in its precision mode at 1 Hz. In-situ ethane ($C_2H_6$) is measured with an Aerodyne Methane/Ethane tunable diode infrared laser direct absorption spectrometer (Yacovitch et al., 2014). There is a 5-10 second time lag in both analyzers that depends on the flow rate and tubing diameter. We use a 1/8" OD (3.175 mm) stainless line for the Picarro (~0.2 slpm flow rate), and a ¼" (6.3 mm) Teflon line for the $CH_4$/ $C_2H_6$ spectrometer (~4 slpm flow rate). This results in lag times of ~5 s for the Aerodyne and ~10 s for the Picarro. The lag time for the Picarro is calculated using a "breath test", whereby we exhale into the air inlet and measure the time required for the $CO_2$ measurement to peak. The ethane lag time is adjusted to maximize the correlation between the ethane and Picarro methane time series in plumes where both gases are emitted. The horizontal wind speed and direction, sampled at 1 Hz, is based on a dual GPS compass that determines aircraft heading and ground speed with sufficient accuracy to resolve the horizontal wind components to about 0.2 m s$^{-1}$ accuracy (Conley et al., 2014). The horizontal wind is calibrated periodically by flying ~5 km L-patterns in the free troposphere; a heading rotation and airspeed adjustment is made to the wind calculation to minimize the dependence of the wind on aircraft heading. These adjustments typically amount to less than 2° rotation and 3% adjustment of the airspeed. In flying the tight circle patterns described below, the pilot does not adjust the rudder trim to maintain the same calibration coefficients in the wind measurement calculation.



**2.2 Large Eddy Simulations**
In order to study the plume behaviour of surface emissions as it relates to sampling in the stacked circles, we
use the LES module of *WRF V3.6.1*. *WRF-LES* explicitly resolves the largest turbulent eddies by filtering the
Navier-Stokes scalar conservation equations at some scale in the inertial subrange, and allowing the smaller
motions beyond the cut-off to be modeled using a sub-grid (also called a subfilter-scale) turbulence
parameterization that is based on properties of the larger-scale, resolved fields. Because the aircraft data is
typically sampled at 1 Hz and the true airspeed is around 70 m s$^{-1}$, we use an LES horizontal grid size roughly
half (40 and 50 m) the distance between aircraft data samples. Because periodic lateral boundary conditions
are imposed on the WRF-LES variables, care must  be taken to ensure that the effluent does not reach the
lateral boundaries of the simulation domain. On the other hand, WRF-LES does not allow for parallelized
computation, making the simulations quite expensive in terms of computation time. We therefore struck a
balance between a large enough domain in horizontal extent (6 and 8 km) such that the effluent would not
reach the downwind boundary before the end of our simulation, while maintaining a grid size small enough to
resolve scales of the aircraft observations. The vertical domain needs to be large enough to encompass a
developing convective boundary layer (CBL), while at the same time containing substantial free tropospheric
flow above to serve as a reservoir that can feed momentum and free-tropospheric scalars to the ABL.
Moreover, the stable region (potential temperature lapse rate $d\theta/dz$ = 5°C/km) between the ABL inversion
base and the top of the domain had to be large enough to damp any wave activity before it could reflect off
the upper boundary and create spurious motions throughout the domain.
The standard *WRF-LES* module is not set up to allow for effluent release, so we implemented a modified
version of the *WRF* source code (*S.-H. Chen, personal communication*) that includes a surface effluent release
with a specified position and release rate. Three different convective simulations were run with varying
resultant mean wind speeds in the boundary layer, and each was allowed 4-5 hours to 'spin-up' dynamically
before the effluent was released at a rate of between 2.9-3.5 kg hr$^{-1}$. The exact release time was selected to
give reasonably stationary ABL depths and turbulent kinetic energy. The conditions for the three simulations
are listed in Table 1, and based on the different wind speeds they span moderate to strongly convective
boundary layers (-$z_i/L$ from ~50 to ~200, where $L$ is the Monin-Obukhov length and $z_i$ is the ABL depth.)
**3 Methods**
**3.1 Theory of Measurement using Gauss's Theorem**
We use an integrated form of the scalar budget equation for a passive, conservative scalar in a turbulent fluid
to estimate the emission of a gas of interest within a cylindrical volume V. The volume is circumscribed by a
series of closed aircraft flight paths (typically circular) flown around the emission source over a range of
altitudes.  The altitudes encompass the lowest safe flight level (usually 75-150 mAGL) up to an altitude where
no discernable change in the trace gas mixing ratio, $\chi$, is observed around the flight loop, $z_{max}$. The scalar in our
case is the mass concentration (i.e., density, $c=\rho_{air}\chi(MW_c/MW_{air})$) of a chemically unreactive species in a
turbulent flow field, $\boldsymbol{u}$ (=$u\boldsymbol{i}+v\boldsymbol{j}+w\boldsymbol{k}$); its Reynolds decomposition is $c = C + c'$, where $C$ is the mean concentration
around each loop and $c'$ is the departure from the loop mean. Figure 1 shows an actual example of the effluent
sampled by the aircraft in a sequence of stacked paths *l* that circumscribe an area, *A*, enclosing the source in a
volume, *V*.  The effluent is carried downwind as it mixes upward in a CBL. A virtual surface circumscribed by
the circular flight tracks is assumed enclosing the source and extending above the vertical extent of the plume





so that there is no vertical transport above that level. To estimate the source strength, we start with the
integral form of the continuity equation:

$$Q_c = \langle \frac{\partial m}{\partial t} \rangle + \iiint \nabla \cdot c\boldsymbol{u} \, dV \qquad (1)$$

where $\langle \, \rangle$ denotes an average over the volume $V$, $Q_c$ is the sum of the internal sources and sinks of $c$ within $V$,
and $m$ is the total mass of $c$ $(m=\langle c \rangle V)$. At this point, we recognize that the flux divergence is composed of two
terms

$$\nabla \cdot c\mathbf{u} = \boldsymbol{u} \cdot \nabla c + c\nabla \cdot \mathbf{u} \qquad (2)$$


In section 3.2 we perform a scale analysis of the terms on the right-hand side (rhs) of equation 2 and show that
the second term, which is proportional to the horizontal wind divergence may be neglected under our normal
flight protocol. This is fortunate because of the difficulty in accurately estimating the horizontal wind
divergence from aircraft measurements (Lenschow et al., 2007). We then decompose the scalar concentration
into a loop mean $C$ and a fluctuating $c'$. The vertical flux across the top (above the plume) and bottom
(ground) are assumed to be zero, leaving us with only the horizontal component, i.e. $c\boldsymbol{u}_h$ where $\boldsymbol{u}_h$ $(= u\mathbf{i}+v\mathbf{j})$.
Since $\nabla C = 0$, equation 2 becomes

$$\boldsymbol{u}_h \cdot \nabla c + c\nabla \cdot \boldsymbol{u}_h = \boldsymbol{u_h} \cdot \nabla(c'). \qquad (3)$$

In order to minimize the contribution from the horizontal divergence term, we remove the loop mean
concentration, $C$. The first term remains unchanged because the gradient of a constant is zero, but the largest
portion of the divergence term is eliminated.

Next, we use Gauss's Theorem to relate the volume integral to a surface integral around the volume that is
sampled by the aircraft flight loops:

$$Q_c = \langle \frac{\partial m}{\partial t} \rangle + \iiint \nabla \cdot (c'\boldsymbol{u}) \, dV = \langle \frac{\partial m}{\partial t} \rangle + \oiint c'\boldsymbol{u} \cdot \hat{\boldsymbol{n}} \, dS \qquad (4)$$

where $S$ is the surface enclosing $V$ and $\hat{n}$ is an outward pointing unit vector normal to the surface.
The surface integral can be broken into three elements: a cylinder extending from the ground up to a level
above significant modification by the emission, the ground surface circumscribed by a low-level (virtual)





circular flight path ($z = 0$), and a nominally horizontal surface circumscribed by a flight path above the level
modified by the source ($z = z_{max}$). We assume there is no significant flux (other than the source of interest) into
or out of the ground,  Then the surface integral is estimated solely from a sequence of closed path integrals
measured by the aircraft at multiple flight levels to estimate the right side of Eq. 5 (blue dashed lines in Fig. 1),

$$\oiint c'\boldsymbol{u} \cdot \hat{\boldsymbol{n}} \, dS = \int_0^{z_{max}} \oint c'\boldsymbol{u_h} \cdot \hat{\boldsymbol{n}} dl \, dz, \tag{5}$$

where $l$ is the flight path.

Combining Eqs. 4 and 5 leads to the result that is the basis for this measurement technique where a series of
horizontal loops are flown around a source region:

$$Q_c = \langle \frac{\partial m}{\partial t} \rangle + \int_0^{z_{max}} \oint c'\boldsymbol{u_h} \cdot \hat{\boldsymbol{n}} dl \, dz \tag{6}$$

Along each path the instantaneous outward flux is computed and summed over the loop to yield the mean flux
divergence via Gauss's Theorem.  A temporal trend of the total mass within the volume ($\frac{\partial m}{\partial t}$) can be estimated
from the flight data and added to the flux divergence integral to obtain the emission rate.

**3.2 Divergence Uncertainty**
In order to estimate the relative error in the horizontal divergence term that we are eliminating, we perform a
scale analysis of the relative size of the two terms that make up the path integral in Eq. 5, using some typical
values of the CBL parameters (a convective velocity scale $w_* = \left( \frac{g}{\Theta_v} \overline{w'\theta_v'} z_i \right)^{1/3} = 1$ m s$^{-1}$, boundary layer
depth, $z_i = 1{,}000$ m, where $g$ is the acceleration due to gravity and $\theta_v$ is the virtual potential temperature) and
sampling geometry (flying at a radius 1 km around the point source). Taylor's (1922) statistical theory of
dispersion in a homogenous and stationary turbulent fluid predicts that the root mean square lateral ($\sigma_y$) and
vertical ($\sigma_z$) dispersion parameters increase linearly with time, or equivalently advection distance, downwind in
the near-field. Weil (1988) shows several examples of the growth of both of these parameters downwind to be
$\sim 0.5 w_*$, which we use here for a rough estimate of a conical plume spreading to quantify the dilution of the
source's emission as it travels downwind to be intercepted by the aircraft. We use a large background mixing
ratio characteristic of global $CH_4$ ($\sim 1.9$ ppmv), estimate the mean gradient by the plume concentration divided
by the distance downwind, and assume a conservatively large horizontal wind divergence of $10^{-5}$ s$^{-1}$, which may
in fact be typical for our small sampling region (Stull, 1988). The results are shown in Figure 2 and, for all but
the smallest sources of a few kg hr$^{-1}$ and wind speeds below 1 m s$^{-1}$, the divergence term is at least an order of
magnitude smaller than the gradient term.



### 3.3 Applying the Theory to the LES Results


We calculated a comparable estimate of $Q_c$ in the LES domain from the air density, concentration, and wind
along circular flight paths as a virtual aircraft would fly. (Willis and Deardorff, 1976) generalized results of their
convection tank experiments to downwind dispersion in the convective boundary layer (CBL) in terms of a
dimensionless length scale $X$, the ratio of the horizontal advection time to the large eddy turnover time:

$$X = \frac{xw_*}{Uz_i} \tag{7}$$

where $x$ is the downwind distance and $U$ is the vertically averaged mean wind speed.
Figure 3 shows the crosswind-integrated concentration profile for the plume release in the UCD50B WRF-LES
run as function of $X$, and normalized height, $Z = z/z_i$. Because of the time limitation due to the periodic
boundary conditions, the plume is averaged for only ~15 minutes of simulation time which is just under a large
eddy turnover time for the conditions of the run. The results displayed in Figure 3 are in good qualitative
agreement with the results of Willis and Deardorff (1976) and Weil et al., (2012) save for the release being at
the surface in our LES study, and at $Z = 0.067$ for the above studies (see Fig. 1 and 2 of Weil et al., (2012)).
Figure 3 shows the maximum concentration being lofted near X~0.2 and leveling off near $Z \sim 0.8$ around $X \sim$
0.6; beyond $X > 1.5$ the plume is fairly well-mixed throughout the extent of the boundary layer.

### 3.4 The Upwind Directed Turbulent Flux


Horizontal turbulent fluxes are generally ignored in boundary layer budget studies due to the fact that while
they are often sizeable in magnitude they do not change significantly over horizontal length scales under
consideration (the horizontal homogeneity assumption). In the vicinity of a point source, however, this is
obviously not a reasonable approximation. Because the method outlined in this work attempts to quantify the
source emission rate through a measured *horizontal* flux, it is worthwhile considering the origins and nature of
these scalar fluxes in turbulent flow. In a wind-tunnel study of flux-gradient relationships Raupach & Legg
(1984) reported that the mean streamwise horizontal heat flux, $UT$, overestimates the total heat flux by
approximately 10% because the turbulent component, $\overline{u'T'}$, is negative; that is, the turbulent flux is upwind,
directed counter to the mean flow. Other researchers have reported an even larger disparity. Field
experiments by Leuning et al. (1985) indicate that the horizontal turbulent flux of a trace gas is ~15% the mean
flux, while Wilson and Shum (1992) suggest it may be 20%, and a recent LES study of particle dispersion over a
plant canopy by Pan et al. [2014] indicates similar magnitudes of 15-20% for the negative turbulent component
of particle fluxes.
To understand why this rather counter-intuitive process occurs it is helpful to inspect the budget equation for
a horizontal scalar flux in a horizontally homogeneous turbulent flow with the x-axis aligned with $U$ [Stull,
1988]:
$$\frac{d\overline{c'u'}}{dt} = -\overline{u'^2}\frac{\partial C}{\partial x} - \overline{u'w'}\frac{\partial C}{\partial z} - \overline{c'w'}\frac{\partial U}{\partial z} - T - \varepsilon \tag{8}$$
where $T$ is a combined 3$^{rd}$ moment and pressure transport term and $\varepsilon$ is dissipation. Because the mean
concentration of $C$ downwind of a source is greater than in the upwind region, the first term is negative, but
decreases in magnitude with distance downwind. Furthermore, the second and third terms are also negative



because the momentum flux, $\overline{u'w'}$, and mean vertical gradient, $\partial C/\partial z$ are negative while the concentration
flux, $\overline{c'w'}$, and wind shear, $\partial U/\partial z$, are positive. However all the terms containing concentrations will tend to
diminish in the downwind direction, so the counter-directed flux will fade with increasing $x$. Based on the
vertical concentration profiles shown in Weil et al. (2012) (their Figures 3 & 4) it can be inferred that the
vertical concentration gradient, $\partial C/\partial z$, changes from negative to positive near $X\sim1$ and becomes negligible for
$X > 2$-$3$. Assuming. Therefore we conclude that when sampling a near surface point source at $X$ less than 2 or 3
it should be necessary to measure the covariance between the concentration and wind because the mean flux
will overestimate the source by somewhere between 10-20%. In this work we use winds and concentrations up
to 1 Hz (Conley et al., 2014), and thus we likely capture a substantial portion of this turbulent flux. Evidence of
this is shown in the cospectra of the outward wind and concentration fluctuation observed in the flight loops in
Figure 4. The simulation results shown are from a $CH_4$ point source with an estimated emission of 46 ±7 kg hr[-1]
which was circled 70 times at a dimensionless radius $X$ of approximately 0.35. All cospectra of sampled sources
have the same structure seen in Figure 4; there is an obvious peak at the mean flight loop frequency (usually
~100 s period) followed by a much smaller negative dip at higher frequencies within the meandering effluent
plume. We believe this to be good evidence that this method captures this important component of the
overall flux away from the source, which cannot be obtained with a traditional mean wind and an integrated
concentration enhancement measurement (White et al., 1976; Ryerson et al., 2001).

**3.5 Choosing the Downwind Sampling Distance**

Determining the optimal sampling distance from the targeted point source is a balance of several factors. First,
not surprisingly, the largest plume signal occurs closest to the source (see Fig. 3). Second, a high degree of
confidence in the results is contingent upon sampling the majority of the plume at and above the lowest flight
altitude, which only occurs downwind after a sufficient time has elapsed to loft the initially near-surface
plume. And third, an attempt is made to sample the plume before it reaches the top of the boundary layer so
that the vertical turbulent entrainment flux does not become appreciable violating the assumption of
negligible flux through the top of the volume V as discussed in Equation 2. Finally, close to the source, the
fluctuations in concentration will be very large, intermittent, at small scales, and highly variable.
To gain further insight into the second feature of the dispersing plume, Figure 5 shows the average horizontal
flux divergence profiles derived from the three WRF-LES runs. Here we discuss a dimensionless $R$, which is
identical to $X$, to emphasize that this scaled downwind distance from the source is a radius of a flight loop. The
flux divergence values are made dimensionless by the boundary layer height, $z_i$, and the source emission rate,
$Q$. Very close to the source, before the plume has had a chance to loft, the flux divergence profile exhibits a
strong gradient below the minimum safe flight altitude, making that term difficult to measure directly, as
shown in Figure 5. Farther from the source, the signal becomes weaker with increasing altitude and eventually
becomes increasingly influenced by entrainment fluxes. We therefore seek a sampling distance that is far
enough to allow sufficient vertical lofting yet close enough so that plume crossings are easily observable
against the background variability and instrument noise, and are not yet influenced by entrainment mixing.
Based on the simulation results presented in Figure 5, we see the gradient below the lowest flight safe altitude
typically becomes very small for $0.4 < R < 0.5$, and therefore we attempt to target that distance to minimize
the extrapolation error from the flight data to the surface. We do not currently measure all the necessary
parameters to estimate R in-flight (namely the heat flux ($\overline{w'\theta'_v}$) which is required to estimate w*. Instead, we



estimate $w_*$ based on the observed boundary layer height, standard deviation of wind speed, and a
parameterization for $w_* = \sigma_u /0.6$ (Caughey and Palmer, 1979).

**3.6 Minimum number of passes**

The atmospheric boundary layer is a turbulent medium, meaning that two passes across a plume at the same
altitude and distance downwind will likely make very different measurements of the trace gases of interest. A
natural question arises as to how many passes are required to develop a statistically sound estimate of the
emission rate. We investigate the number of passes required to obtain a statistically robust estimate using the
WRF-LES results and a controlled release experiment. By calculating the horizontal flux divergences with a
virtual airplane flying through the simulated tracer field, and then randomly sampling the flux divergences
from each of the legs and plotting the resultant estimated emission rate as a function of the number of
samples used we obtain the results presented in Figure 6. The gray region around the red line mean represents
the standard deviation of estimates based on a random set of loops. Figure 7 shows similar results from an
analysis of actual flight data from the ethane controlled release test near Denver, Colorado on November 19,
2014. It is evident from both the simulation data and the field data that a statistically stable and accurate
emission estimate seems to be achieved somewhere between 20-25 loops around the source.

**3.7 Discretization and Altitude Binning the Flux Divergence Data**

Measurements of the relevant scalars (e.g. $CH_4$) and meteorological variables are not continuous, but reported
on discrete time intervals.  For our analyses, we interpolate all measurements including GPS (3 Hz), methane (1
Hz), and temperature (1 Hz), horizontal wind (1 Hz), air density (1 Hz) onto a synchronous 1 Hz time series.
Next, we estimate the path integral for each individual loop of the flux normal to the flight path by summing
up the flux contributions times the sample length around each loop and then summing over the height
intervals,

$$Q_c = \left\langle \frac{\partial m}{\partial t} \right\rangle + \oiint \boldsymbol{F_c} \cdot \widehat{\boldsymbol{n}} \, dS = \frac{\Delta m}{\Delta t} + \sum_{z=0}^{z=Z_t} \left( \sum_0^L (\rho \cdot u_n) \cdot \Delta s \right) \cdot \Delta z, \tag{9}$$

where $\rho$ is the scalar air density, $u_n$ is the wind speed normal to the flight path, $\Delta s$ is the distance covered
during the 1 s time interval of each measurement and $L$ is the distance covered in one complete circuit.  The
outer summation sums each of the discrete vertical laps from the bottom (z = 0) to the highest lap (z = $z_t$).  If all
laps were sampled at equidistant altitudes, the total divergence could be calculated as the average divergence
of all laps multiplied by the top altitude. However, because there is greater variability at lower altitudes, as
demonstrated by the widening standard deviations in the theoretical flux divergence profiles shown in Figure
4, more sampling laps are required at lower altitudes to increase the statistical validity of the largest flux
divergence values. To ensure that all altitudes are nearly equally weighted, we divide the vertical range into six
equally spaced bins, save for the lowest one which is extrapolated to the surface, and then average the
measurements from the laps within each bin. The total emission is the sum of the flux in each bin multiplied by
the bin width. We also performed 6 flights where we sampled equally at all altitudes to allow a comparison of
the direct average versus the binned results, and in all of these flights the values derived by the two methods
agreed to within 5%.



**3.8 Error Analysis**

Our method assumes a temporally constant emission source and that the plume is in steady-state during each individual measurement interval. The leg-to-leg variability is primarily driven by the stochastic nature of turbulence (e.g. we sample the plume on one lap, miss it on another). By aggregating the laps into vertical bins, we can use the standard deviation of the horizontal fluxes within each bin as an estimate of the uncertainty within that bin. Then the total uncertainty in the estimate of the flux divergence is simply estimated by adding up the individual bin uncertainties in quadrature.

The first term on the rhs of Equation 6 is the time rate of change of the scalar mass within the plume. This is estimated by performing a least squares fit, using time and altitude as the predictors and the trace gas mixing ratio as the response variable, assuming the cylinder volume does not change, i.e.:

$$c = \alpha t + \beta z \qquad (7)$$

The first coefficient ($\alpha$) is the time rate of change, and the uncertainty in $\alpha$ (usually expressed in units of ppb hr$^{-1}$) is then converted to density units using the average pressure and temperature measured around the cylinder and multiplied by the volume of the cylinder to obtain the uncertainty in the rate of change of the total scalar mass within the cylinder.

Finally, the two uncertainties (time rate of change and flux divergence) are combined in quadrature to obtain the total uncertainty in the flux estimate.

**4. Results and Discussion**

We use measurements from three sets of flights to characterize the accuracy of this estimation method. We flew 4 days measuring a natural gas controlled release provided by the Pacific Gas & Electric Company (PG&E), 2 days measuring an ethane controlled release provided by Aerodyne Research, Inc., and 6 power-plant flights where our estimates are compared with reported hourly power plant $CO_2$ emissions.

**4.1 Ethane Controlled Releases**

Two experiments with known/controlled ethane releases were performed in collaboration with the Aerodyne Mobile Laboratory team. Pure ethane was released and measured with a flowmeter by the Aerodyne ground crew. The Colorado site (November, 2014) was in a remote area approximately 105 miles NE of Denver. This site was chosen because of the flat terrain and lack of other nearby ethane sources that could pollute the controlled release plume. Agreement was excellent, with the estimated emission just 2% over the actual controlled release rate. The second Aerodyne controlled release in Arkansas on October 3, 2015 was



performed at a site surrounded by nearby emission sources and an elevation change (~70m) within the aircraft
flight path. The aircraft derived ethane emission estimate was 25% higher than the controlled release rate and
the calculated uncertainty was significantly higher than on other sites (Table 2).
A significant upwind ethane source was observed during the Arkansas experiment. This source was evident on
roughly half of the upwind passes, suggesting that techniques which rely on a limited number of upwind
passes to characterize the background could have a large random error and thus erroneously estimate the
upwind source strength.  A similar problem would affect those techniques that employ a downwind transect,
using the edges of that transect to estimate the background concentration.  These observations demonstrate
the complication (and bias) that can arise from nearby sources.  Since this method integrates all the emission
sources in the area within the flight circle and a small distance upwind of the circle depending on the vertical
mixing, estimates from the Gauss's method should be biased high if there are sources within that area.  The
average error between the two measurements is 13%.

**4.2 Natural Gas Controlled Releases**

In conjunction with PG&E, we performed two sets of two-day ground-level controlled release experiments
from existing PG&E facilities, exactly one year apart.  The first set was performed southeast of Sacramento
near the town of Rio Vista, CA at the Rio Vista "Y" station and the second set near Bakersfield, CA.  For the Rio
Vista test, the release rate was not calibrated with a flow meter but, based on the size of the orifice and the
upstream pressure, the release rate was estimated at 15.2 kg hr$^{-1}$.  This release rate is an estimate of the total
gas being released which is a combination of primarily $CH_4$ and $C_2H_6$.  We use the regression fit of ethane to
methane (averaging 0.085 by mass) to estimate the actual release rate of each scalar.  The Bakersfield test
results were compromised (release rate not recorded) and are not included here.
In comparison with the $C_2H_6$ controlled release, $CH_4$ controlled releases suffer from the effect of small
enhancements relative to the background. During the Bakersfield release, the largest enhancement that we
measured was 100 ppb, with 30-40 ppb being typical.  Using a typical background level of 1.9 ppm, a 40 ppb
enhancement represents 2% of the background.  In contrast, for ethane the enhancements are as large or
larger than the background.   The results of the methane controlled release tests are shown in Table 3  and
indicate aircraft derived estimate within 17% of the controlled release rate.  This large background results in
increased uncertainty in the emission calculation.  The average difference between the estimated emission
and the calculated flow rate is 13%.

**4.3 Power Plant Flights**

Power plants in the U.S. are required to report $CO_2$ emissions to EPA (https://ampd.epa.gov/ampd) on an
hourly basis. The accuracy of the reported $CO_2$ emissions has been determined to be ±10.8-11.0% based on
reported U.S. average differences between Energy Information Administration (EIA) fuel-based estimates and
EPA continuous emission monitoring based estimates (Ackerman and Sundquist, 2008; Peischl et al., 2010;
Quick, 2014).  Also, Peischl et al. (2010) determined an accuracy of power plants reporting $CO_2$ emissions in
Texas of ±14.0% based on differences between observed downwind $SO2/CO_2$ and $NOx/CO_2$ emission ratios and
those reported via EPA continuous emission monitoring (Peischl et al., 2010). Here, we use the slightly larger



uncertainty from Peischl et al. (2010). An additional uncertainty arises from temporal emission variability
(hourly average reported $CO_2$ emissions vs. <1 hour power plant flights that may cover parts of two reported
consecutive hourly values). We estimate the total reported uncertainty by summing in quadrature the Peischl
estimate and the relative difference between two reported consecutive hourly $CO_2$ emission values closest to
the time of the power plant sampling. The aircraft frequently encountered power plants during oil & gas
monitoring campaigns, but usually did not have the flight time to perform a full emissions characterization of
the power plant.  Here we limit our comparison to days when the aircraft performed a minimum of 10 laps
around the plant, thus excluding the quick fly-bys where uncertainties would be unacceptably large.  The
results are presented in Table 4 and indicate very good agreement between Gauss's method and the reported
$CO_2$ emissions with the averaged magnitude of the difference being 10.6%.  Power plant emissions are "hot"
gases and very buoyant, in contrast to a surface emission source that is cooled by expansion and therefore not
buoyant.  The average difference between the reported and measured emissions for the 5 power plants is
407   11%.


In addition to the controlled release experiments, hundreds of sites have been measured using this technique
with varying levels of success.  Ideal conditions include flat terrain, ample sunlight to promote vertical mixing,
consistent winds, and no nearby competing sources.  With these conditions, measured uncertainties are quite
low, often better than 10%. As the conditions deteriorate from the ideal to situations involving complex
terrain, variable winds or nearby upwind sources, measured uncertainties can exceed several hundred percent.
Without sufficient vertical mixing, trace gases emitted at the surface may never reach the minimum safe flight
altitude, rendering this technique completely unsuitable.



| Simulation | $\Delta x, \Delta y$ (m) | Lx, Ly (km) | $\Delta Z$ (km) | $\Delta t$ (s) | $\Delta z$ (m) | ABL Depth (m) | ABL mean wind (ms$^{-1}$) | $w_*$ (ms$^{-1}$) | $-z_i/L$ | $X_{max}$ |
|---|---|---|---|---|---|---|---|---|---|---|
| **UCD50A** | 50 | 8 | 2.5 | 0.30 | 8? | 750 | 2 | 0.92 | 210 | 4.5 |
| **UCD50B** | 50 | 8 | 2.5 | 0.30 | 8? | 600 | 3.8 | 0.86 | 73 | 3.6 |
| **UCD40** | 40 | 6 | 2.5 | 0.24 | 10 | 850 | 4.5 | 0.96 | 53 | 2.4 |

**Table 1 -** **Domain and micrometeorological parameters for the three WRF-LES experiments in this study.** *L* **represents the Monin-Obukhov length.**

| Experiment Location | Date | Laps | Released $CH_4$ kg hr$^{-11}$ | Estimated $CH_4$ kg hr$^{-1}$ | Released $C_2H_6$ kg hr$^{-1}$ | Estimated $C_2H_6$ kg hr$^{-1}$ | Difference |
|---|---|---|---|---|---|---|---|
| Colorado | 11/19/14 | 50 | 0.0 | -0.1±0.3 | 5.5±0.5 | 5.6±2.9 | +2% |
| Arkansas | 10/03/15 | 19 | 0.0 | -3.4±12.3 | 8.1±0.8 | 10.0±6.1 | +24% |

**Table 2 - Ethane controlled releases.**

| Experiment Location | Date | Laps | Released $CH_4$ kg hr$^{-1}$ | Estimated $CH_4$ kg hr$^{-1}$ | Released $C_2H_6$ kg hr$^{-1}$ | Estimated $C_2H_6$ kg hr$^{-1}$ | Difference |
|---|---|---|---|---|---|---|---|
| Rio Vista | 11/03/14 | 37 | 13.9±2.8 | 12.8±8.5 | 1.2±0.5 | 0.6±0.4 | -8% |
| Rio Vista | 11/04/14 | 27 | 13.9±2.8 | 11.5±3.2 | 1.2±0.5 | 0.5±0.3 | -17% |

**Table 3 - Natural Gas controlled release**



| Power Plant | Date | Hour UTC | Laps | Reported $CO_2$ T hr$^{-1}$ | Estimated $CO_2$ T hr$^{-1}$ | Difference |
|---|---|---|---|---|---|---|
| Rocky Mountain Energy | 10/06/14 | 20 | 19 | 99±14 | 111±24 | 13% |
| Saint Vrain | 10/04/14 | 19 | 21 | 124±17 | 122±41 | -1% |
| Pawnee | 11/19/14 | 20 | 14 | 575±81 | 555±160 | -3% |
| Saint Vrain | 09/17/15 | 20 | 14 | 361±54 | 280±115 | -23% |
| Four Corners Power Plant | 04/11/15 | 18 | 12 | 1289±387 | 1119±343 | -13% |

**Table 4 - Power Plant estimates**

**4. Acknowledgements**

*Funding for this study provided by the California Energy Commission (CEC) and the US Department of Energy (DOE).*

*Funding for the Denver and Arkansas portion of this work was provided by RPSEA/NETL contract no 12122-95/DE-AC26-07NT42677 to the Colorado School of Mines. Cost share was provided by Colorado Energy Research Collaboratory, the National Oceanic and Atmospheric Administration Climate Program Office, the National Science Foundation (CBET-1240584), Southwestern Energy, XTO, Chevron, Statoil and the American Gas Association, many of whom also provided operational data and/or site access. We also thank Professor Shuhua Chen for assistance with the WRF-LES code modifications and advice. The National Center for Atmospheric Research is sponsored by the National Science Foundation. This work was supported in part by the NOAA AC4 program under grant NA14OAR0110139 and the Bureau of Land Management, Grant L15PG00058*





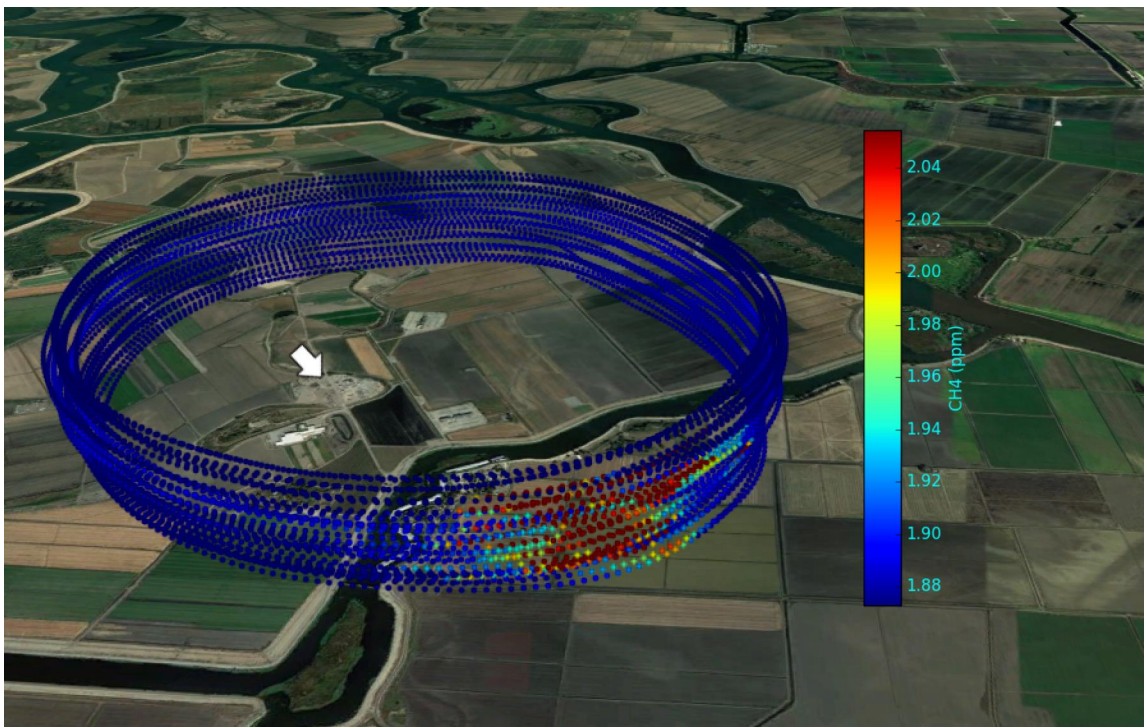

**Figure 1 - Map of the airplane flight pattern sampling a methane plume emanating from an underground storage facility. Wind direction is indicated by the white arrow and the methane mixing ratio is given by the color bar to right**

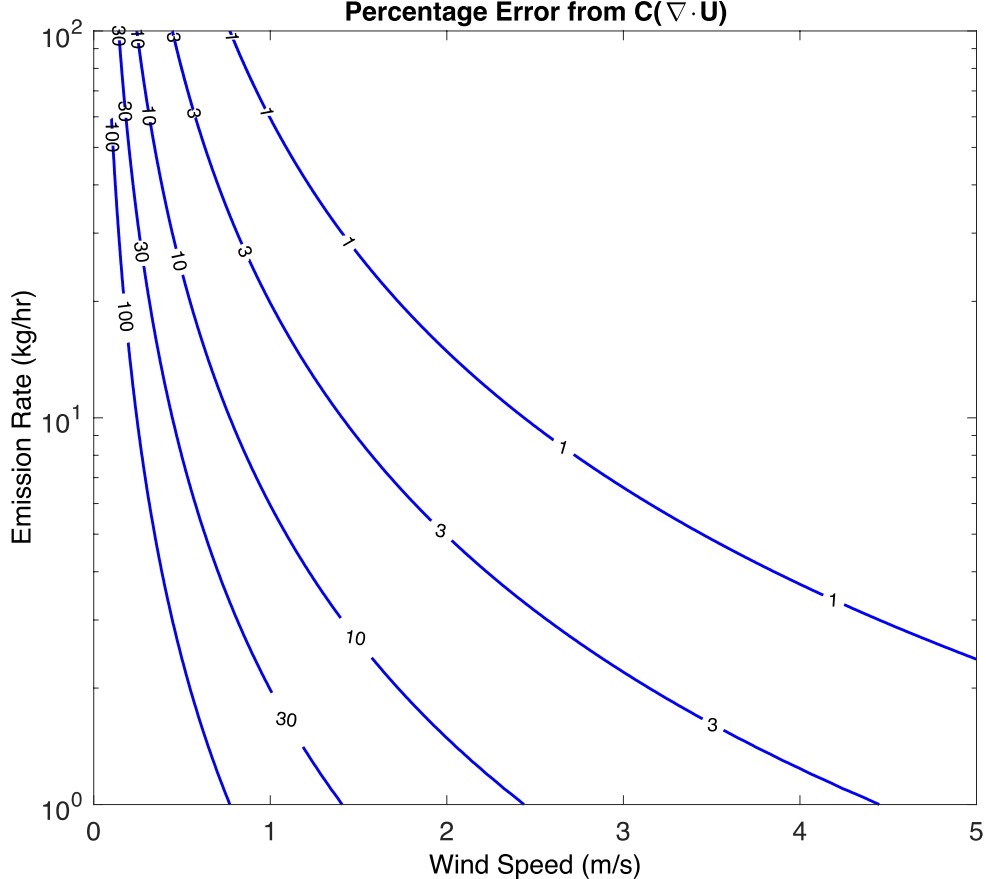

Figure 2 - Graphical representation of the relative magnitude (%) of the contribution of the horizontal wind divergence to the horizontal advective terms in Equation 4, as a function of wind speed and source magnitude for methane, using a typical global background of 1.9 ppm and divergence of $10^{-5}$ $s^{-1}$.





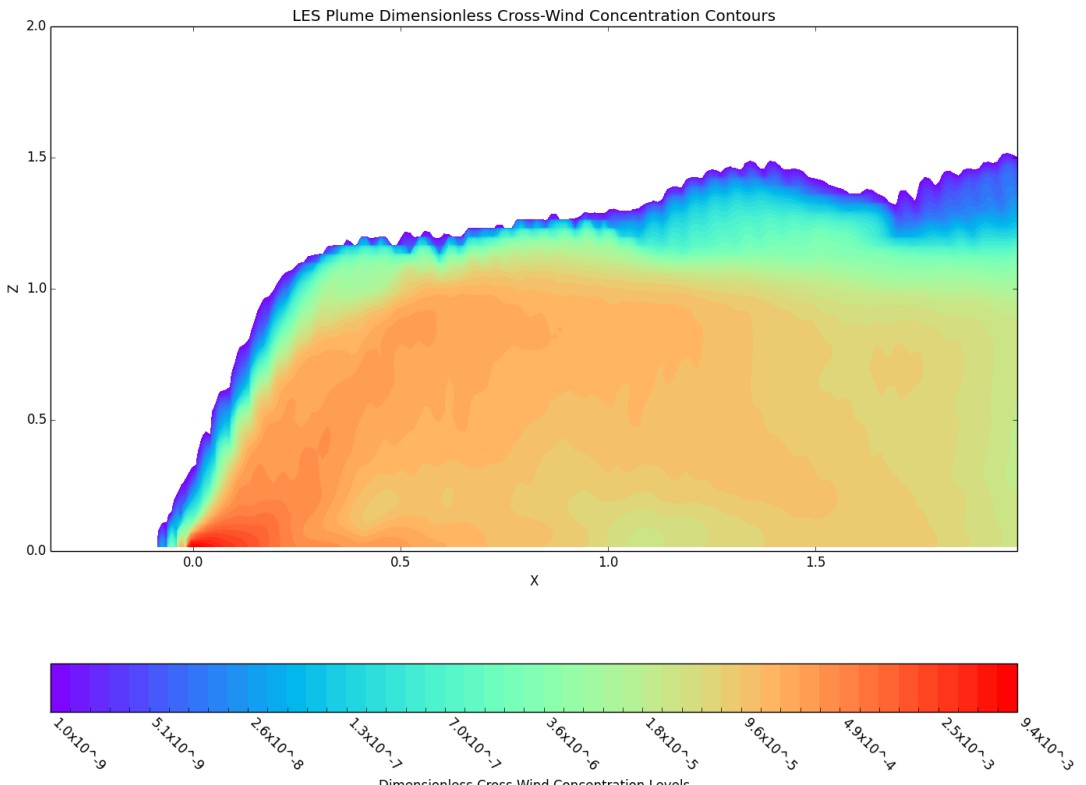

**Figure 3 Cross wind integrated concentrations of an effluent plume released at the surface in the UCD50B simulation. The data are averaged over 15 minutes of simulation time.**

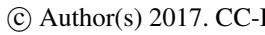


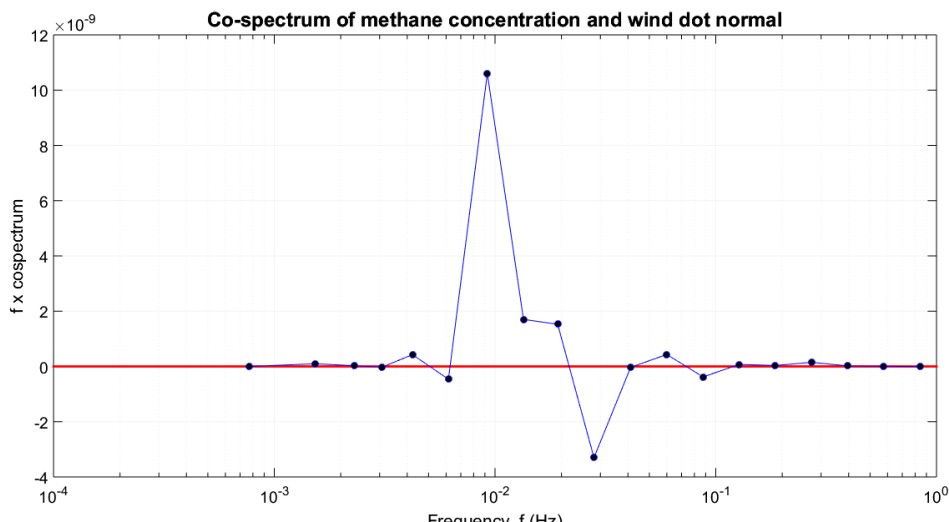

**Figure 4 Average cospectrum of the outward directed component of the observed wind and the methane concentration from 70 laps around a point source near San Antonio, Texas. The peak at 10$^{-2}$ Hz corresponds to the period of the circle.**




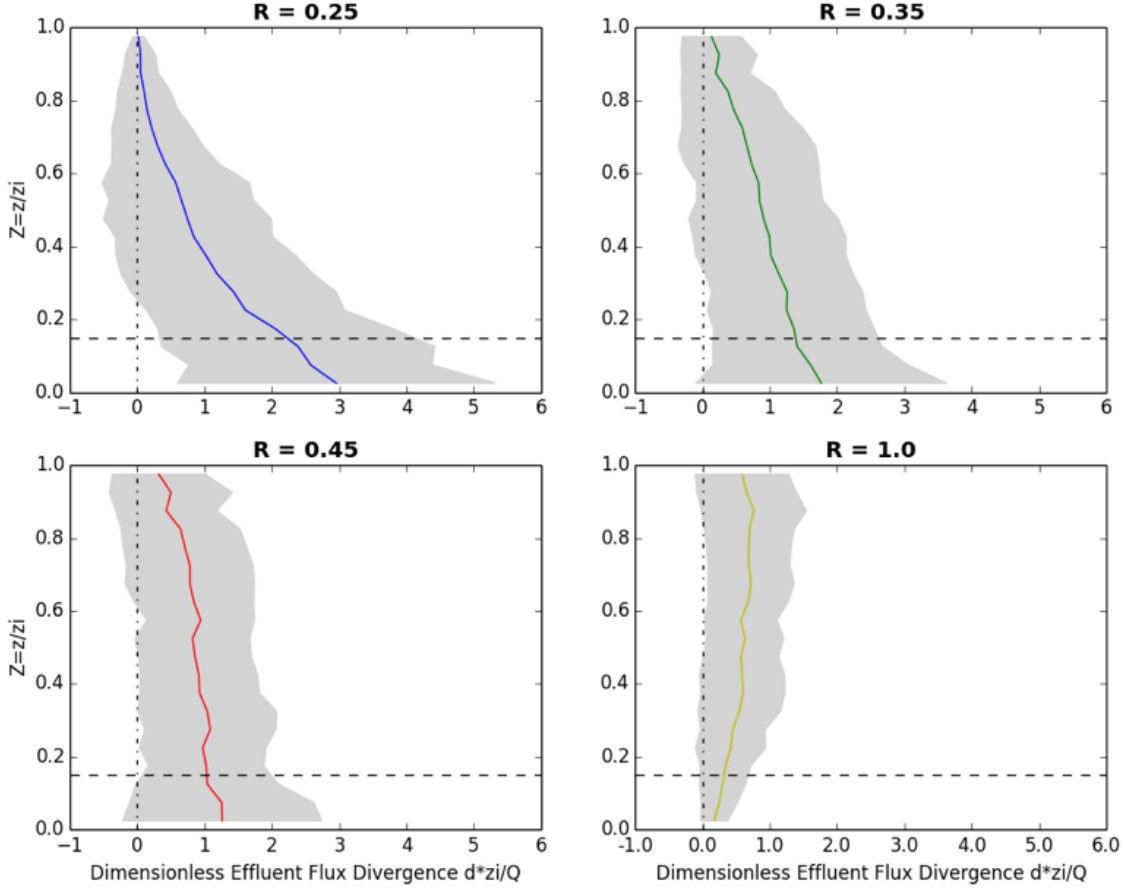

**Figure 5 Dimensionless flux divergence profiles generated from averaging over 3 different WRF-LES runs using 30 time steps for each one. The horizontal flux divergences (d) are normalized by the boundary layer height, $z_i$, and source strength, $Q$. The colored profiles are averages at various dimensionless distances, $R$=0.1, 0.2, 0.3, and 0.4 and the gray areas represent one standard deviation about the mean. The horizontal dashed lines are the approximate lowest safe flight altitude.**



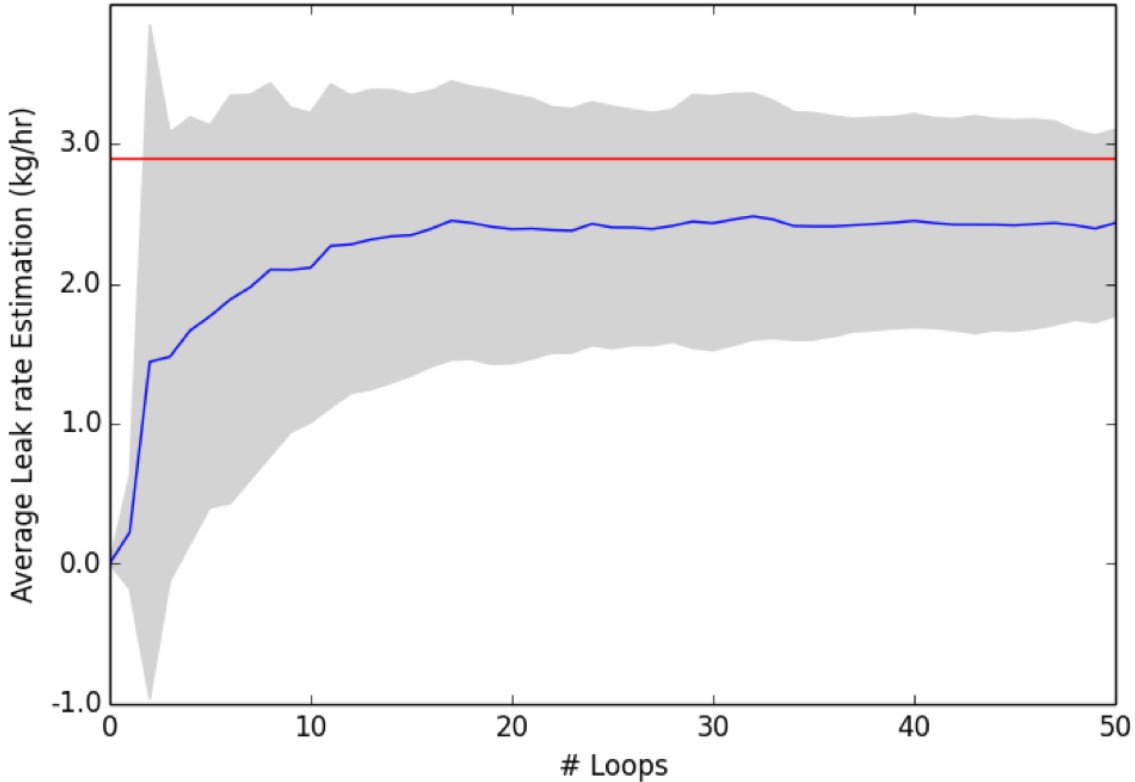

**Figure 6-CASE UCD50B: This figure shows the rate of convergence toward the final leak rate estimation, and shows that by around 15 laps, the emissions estimate (blue line) has stabilized to 2.5 kg hr$^{-1}$ compared to the actual leak rate (red line) of 2.9 kg hr$^{-1}$. Dimensionless distance R = 0.25, 50 realizations. Grey area represents 1 standard deviation.**

.





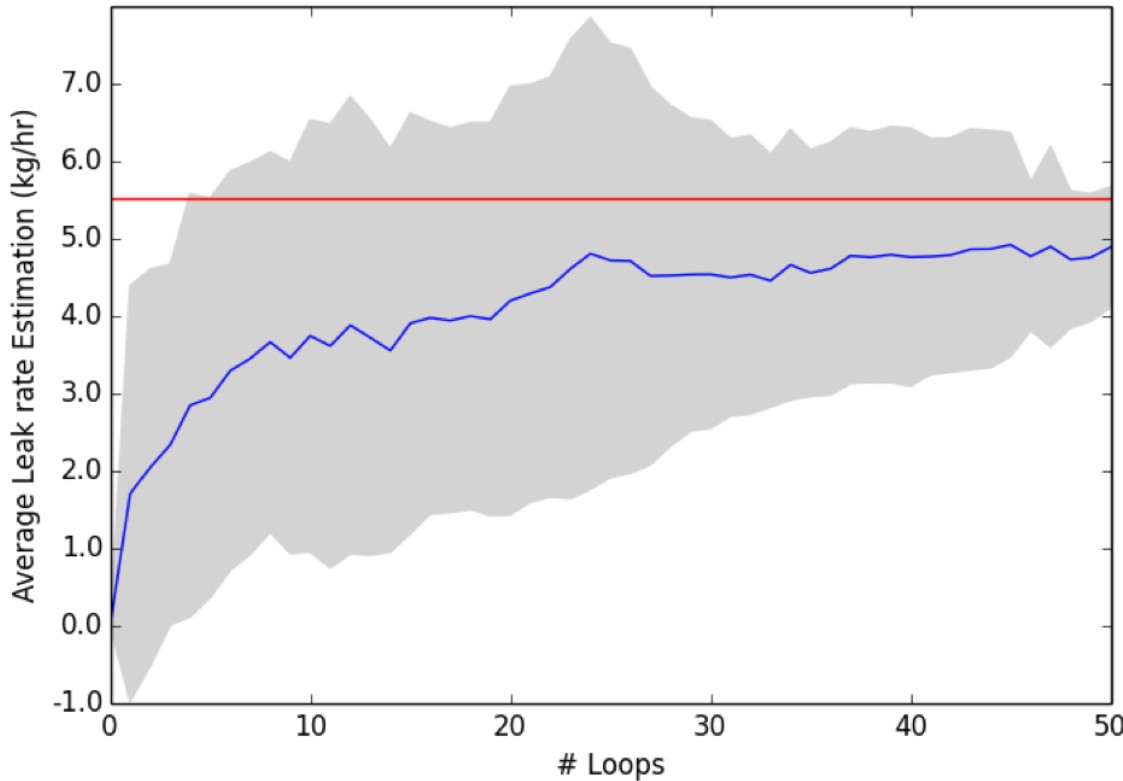

**Figure 7 Case Aerodyne: Average Leak Rate Estimation.  This leak shows a slightly higher number of laps before convergence (~25 laps).   This simulation was performed using the Aerodyne controlled release near Denver, Colorado on November 19, 2014.**




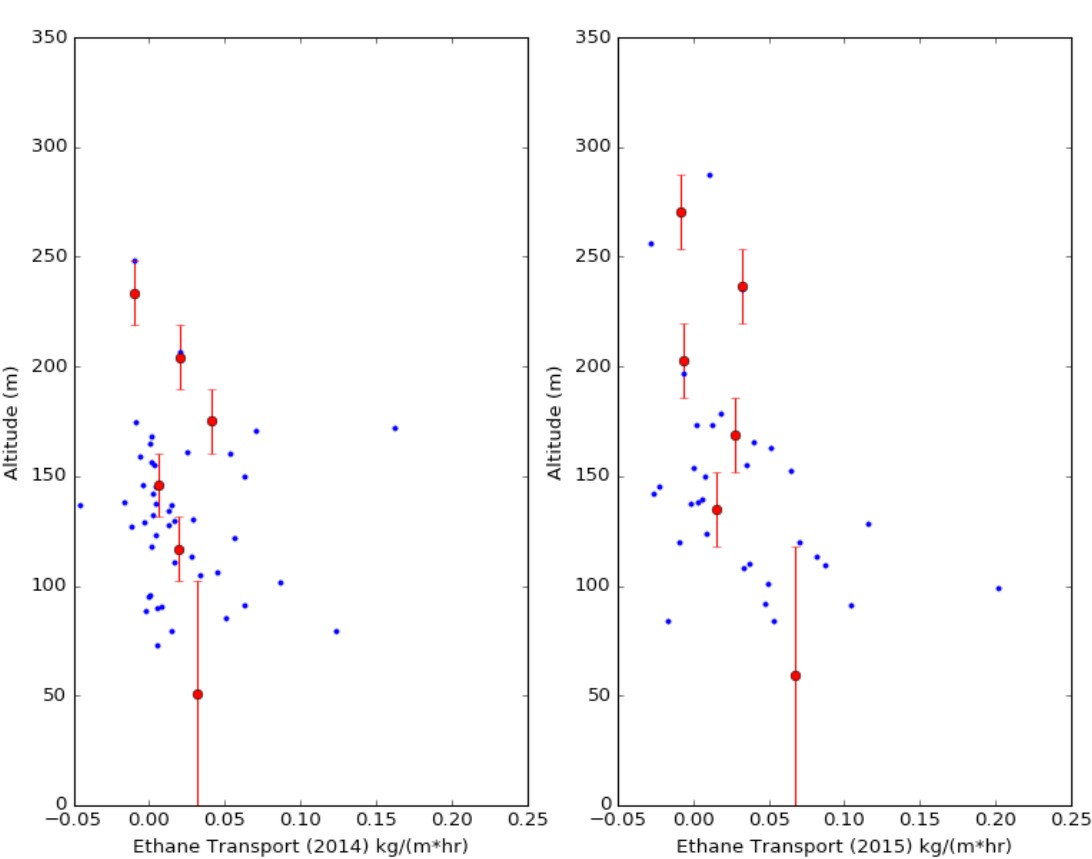

**Figure 8 - Ethane flux divergence profiles for Aerodyne controlled releases near Denver, Colorado on November 19, 2014 (left) and in Bee Branch, Arkansas on October 3, 2015 (right). Blue dots represent individual flight loop measurements and the red circles represent the bin average values for altitude intervals represented by the red bars. The dimensionless distance downwind for each is approximately 0.43 and 0.24, respectively.**





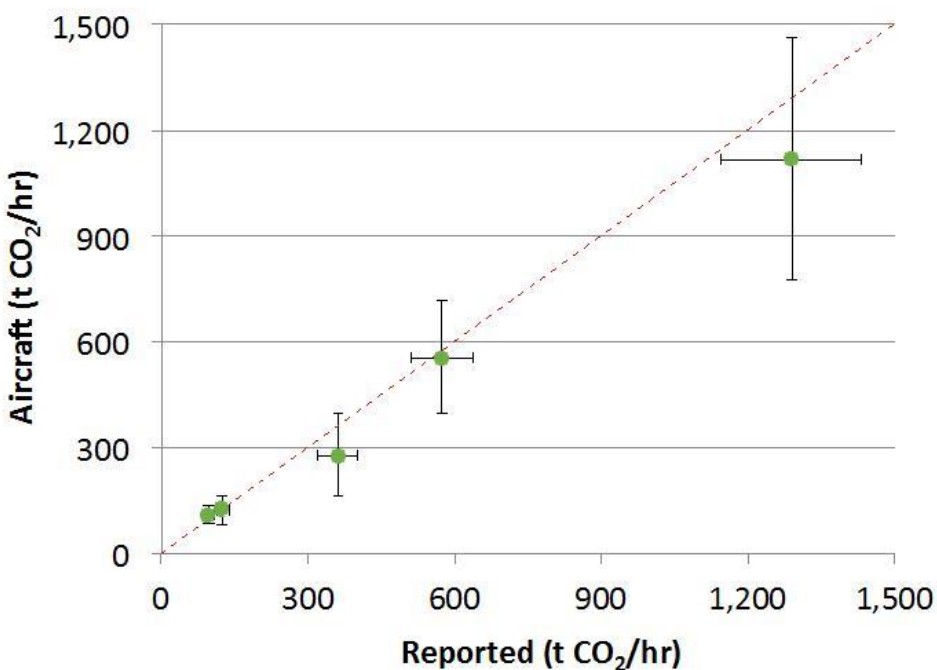

**Figure 9 Comparison of aircraft versus reported power plant emissions.**

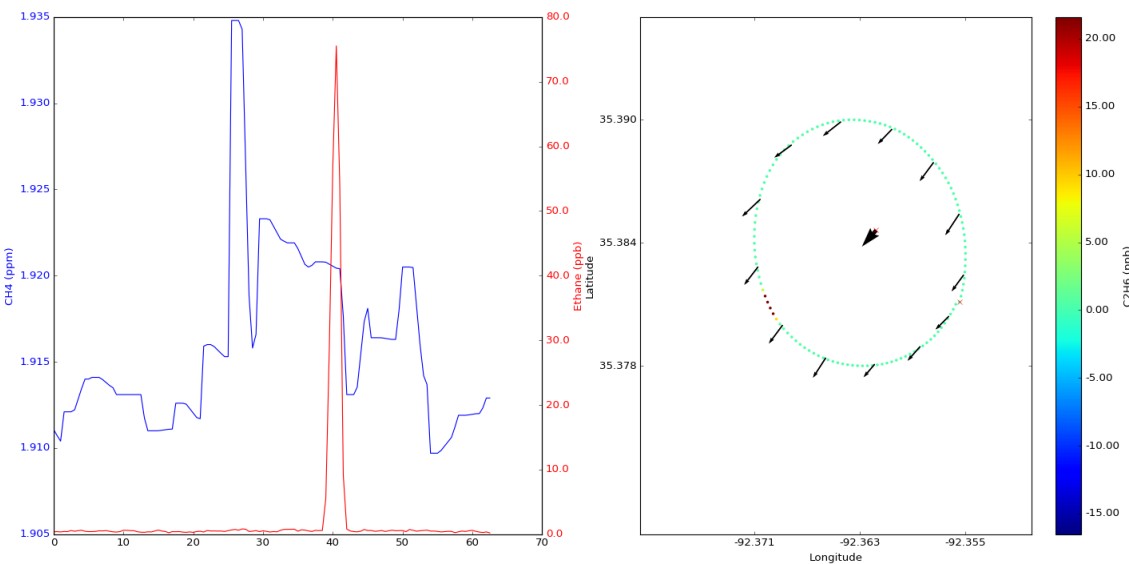

**Figure 10 Time series of methane and ethane (left plot) along with geographic distribution of methane during the second ethane controlled release.**





Ackerman, K. V. and Sundquist, E. T.: Comparison of two US power-plant carbon dioxide emissions data sets, Environmental Science & Technology, 42, 5688-5693, 2008.

Alfieri, J. G. and Blanken, P. D.: How representative is a point? The spatial variability of flux measurements across short distances. In: Remote Sensing and Hydrology, Neale, C. M. U. and Cosh, M. H. (Eds.), IAHS Publication, 2012.

Alfieri, J. G., Kustas, W. P., Prueger, J. H., Hipps, L. E., Evett, S. R., Basara, J. B., Neale, C. M. U., French, A. N., Colaizzi, P., Agam, N., Cosh, M. H., Chavez, J. L., and Howell, T. A.: On the discrepancy between eddy covariance and lysimetry-based surface flux measurements under strongly advective conditions, Advances in Water Resources, 50, 62-78, 2012.

Bergamaschi, P., Krol, M., Dentener, F., Vermeulen, A., Meinhardt, F., Graul, R., Ramonet, M., Peters, W., and Dlugokencky, E. J.: Inverse modelling of national and European CH4 emissions using the atmospheric zoom model TM5, Atmospheric Chemistry and Physics, 5, 2431-2460, 2005.

Beswick, K. M., Simpson, T. W., Fowler, D., Choularton, T. W., Gallagher, M. W., Hargreaves, K. J., Sutton, M. A., and Kaye, A.: Methane emissions on large scales, Atmospheric Environment, 32, 3283-3291, 1998.

Caughey, S. J. and Palmer, S. G.: SOME ASPECTS OF TURBULENCE STRUCTURE THROUGH THE DEPTH OF THE CONVECTIVE BOUNDARY-LAYER, Quarterly Journal of the Royal Meteorological Society, 105, 811-827, 1979.

Caulton, D. R., Shepson, P. B., Santoro, R. L., Sparks, J. P., Howarth, R. W., Ingraffea, A. R., Cambaliza, M. O. L., Sweeney, C., Karion, A., Davis, K. J., Stirm, B. H., Montzka, S. A., and Miller, B. R.: Toward a better understanding and quantification of methane emissions from shale gas development, Proceedings of the National Academy of Sciences, 111, 6237-6242, 2014.

Chang, R. Y. W., Miller, C. E., Dinardo, S. J., Karion, A., Sweeney, C., Daube, B. C., Henderson, J. M., Mountain, M. E., Eluszkiewicz, J., Miller, J. B., Bruhwiler, L. M. P., and Wofsy, S. C.: Methane emissions from Alaska in 2012 from CARVE airborne observations, Proceedings of the National Academy of Sciences of the United States of America, 111, 16694-16699, 2014.

Conley, S., Franco, G., Faloona, I., Blake, D. R., Peischl, J., and Ryerson, T. B.: Methane emissions from the 2015 Aliso Canyon blowout in Los Angeles, CA, Science, 351, 1317-1320, 2016.

Conley, S. A., Faloona, I. C., Lenschow, D. H., Karion, A., and Sweeney, C.: A Low-Cost System for Measuring Horizontal Winds from Single-Engine Aircraft, Journal of Atmospheric and Oceanic Technology, 31, 1312-1320, 2014.

Crosson, E. R.: A cavity ring-down analyzer for measuring atmospheric levels of methane, carbon dioxide, and water vapor, Applied Physics B-Lasers and Optics, 92, 403-408, 2008.



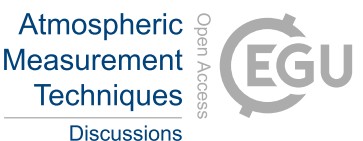

Czepiel, P. M., Mosher, B., Harriss, R. C., Shorter, J. H., McManus, J. B., Kolb, C. E., Allwine, E., and Lamb, B. K.: Landfill methane emissions measured by enclosure and atmospheric tracer methods, Journal of Geophysical Research-Atmospheres, 101, 16711-16719, 1996.

Denmead, O. T., Harper, L. A., Freney, J. R., Griffith, D. W. T., Leuning, R., and Sharpe, R. R.: A mass balance method for non-intrusive measurements of surface-air trace gas exchange, Atmospheric Environment, 32, 3679-3688, 1998.

Gallagher, M. W., Choularton, T. W., Bower, K. N., Stromberg, I. M., Beswick, K. M., Fowler, D., and Hargreaves, K. J.: MEASUREMENTS OF METHANE FLUXES ON THE LANDSCAPE SCALE FROM A WETLAND AREA IN NORTH SCOTLAND, Atmospheric Environment, 28, 2421-2430, 1994.

Gerbig, C., Lin, J. C., Wofsy, S. C., Daube, B. C., Andrews, A. E., Stephens, B. B., Bakwin, P. S., and Grainger, C. A.: Toward constraining regional-scale fluxes of CO2 with atmospheric observations over a continent: 2. Analysis of COBRA data using a receptor-oriented framework, Journal of Geophysical Research-Atmospheres, 108, 27, 2003.

Gordon, M., Li, S. M., Staebler, R., Darlington, A., Hayden, K., O'Brien, J., and Wolde, M.: Determining air pollutant emission rates based on mass balance using airborne measurement data over the Alberta oil sands operations, Atmos. Meas. Tech., 8, 3745-3765, 2015.

Hacker, J. M., Chen, D. L., Bai, M., Ewenz, C., Junkermann, W., Lieff, W., McManus, B., Neininger, B., Sun, J. L., Coates, T., Denmead, T., Flesch, T., McGinn, S., and Hill, J.: Using airborne technology to quantify and apportion emissions of CH4 and NH3 from feedlots, Animal Production Science, 56, 190-203, 2016.

Hiller, R. V., Neininger, B., Brunner, D., Gerbig, C., Bretscher, D., Kunzle, T., Buchmann, N., and Eugster, W.: Aircraft-based CH4 flux estimates for validation of emissions from an agriculturally dominated area in Switzerland, Journal of Geophysical Research-Atmospheres, 119, 4874-4887, 2014.

Hirsch, A. I., Michalak, A. M., Bruhwiler, L. M., Peters, W., Dlugokencky, E. J., and Tans, P. P.: Inverse modeling estimates of the global nitrous oxide surface flux from 1998-2001, Global Biogeochemical Cycles, 20, 2006.

Kalthoff, N., Corsmeier, U., Schmidt, K., Kottmeier, C., Fiedler, F., Habram, M., and Slemr, F.: Emissions of the city of Augsburg determined using the mass balance method, Atmospheric Environment, 36, S19-S31, 2002.

Karion, A., Sweeney, C., Kort, E. A., Shepson, P. B., Brewer, A., Cambaliza, M., Conley, S. A., Davis, K., Deng, A. J., Hardesty, M., Herndon, S. C., Lauvaux, T., Lavoie, T., Lyon, D., Newberger, T., Petron, G., Rella, C., Smith, M., Wolter, S., Yacovitch, T. I., and Tans, P.: Aircraft-Based Estimate of Total Methane Emissions from the Barnett Shale Region, Environmental Science & Technology, 49, 8124-8131, 2015.

Karion, A., Sweeney, C., Petron, G., Frost, G., Hardesty, R. M., Kofler, J., Miller, B. R., Newberger, T., Wolter, S., Banta, R., Brewer, A., Dlugokencky, E., Lang, P., Montzka, S. A., Schnell, R., Tans, P., Trainer, M., Zamora, R., and Conley, S.: Methane emissions estimate from airborne measurements over a western United States natural gas field, Geophysical Research Letters, 40, 4393-4397, 2013.





Lamb, B. K., McManus, J. B., Shorter, J. H., Kolb, C. E., Mosher, B., Harriss, R. C., Allwine, E., Blaha, D., Howard, T., Guenther, A., Lott, R. A., Siverson, R., Westberg, H., and Zimmerman, P.: DEVELOPMENT OF ATMOSPHERIC TRACER METHODS TO MEASURE METHANE EMISSIONS FROM NATURAL-GAS FACILITIES AND URBAN AREAS, Environmental Science & Technology, 29, 1468-1479, 1995.

Lavoie, T. N., Shepson, P. B., Cambaliza, M. O. L., Stirm, B. H., Karion, A., Sweeney, C., Yacovitch, T. I., Herndon, S. C., Lan, X., and Lyon, D.: Aircraft-Based Measurements of Point Source Methane Emissions in the Barnett Shale Basin, Environmental Science & Technology, 49, 7904-7913, 2015.

Lenschow, D. H., Savic-Jovcic, V., and Stevens, B.: Divergence and vorticity from aircraft air motion measurements, Journal of Atmospheric and Oceanic Technology, 24, 2062-2072, 2007.

Leuning, R., Freney, J. R., Denmead, O. T., and Simpson, J. R.: A SAMPLER FOR MEASURING ATMOSPHERIC AMMONIA FLUX, Atmospheric Environment, 19, 1117-1124, 1985.

Mays, K. L., Shepson, P. B., Stirm, B. H., Karion, A., Sweeney, C., and Gurney, K. R.: Aircraft-Based Measurements of the Carbon Footprint of Indianapolis, Environmental Science & Technology, 43, 7816-7823, 2009.

Miller, J. B., Lehman, S. J., Montzka, S. A., Sweeney, C., Miller, B. R., Karion, A., Wolak, C., Dlugokencky, E. J., Southon, J., Turnbull, J. C., and Tans, P. P.: Linking emissions of fossil fuel CO2 and other anthropogenic trace gases using atmospheric (CO2)-C-14, Journal of Geophysical Research-Atmospheres, 117, 23, 2012.

Miller, S. M., Wofsy, S. C., Michalak, A. M., Kort, E. A., Andrews, A. E., Biraud, S. C., Dlugokencky, E. J., Eluszkiewicz, J., Fischer, M. L., Janssens-Maenhout, G., Miller, B. R., Miller, J. B., Montzka, S. A., Nehrkorn, T., and Sweeney, C.: Anthropogenic emissions of methane in the United States, Proceedings of the National Academy of Sciences of the United States of America, 110, 20018-20022, 2013.

Muhle, S., Balsam, I., and Cheeseman, C. R.: Comparison of carbon emissions associated with municipal solid waste management in Germany and the UK, Resour. Conserv. Recycl., 54, 793-801, 2010.

Neef, L., van Weele, M., and van Velthoven, P.: Optimal estimation of the present-day global methane budget, Global Biogeochemical Cycles, 24, 2010.

Nisbet, E. and Weiss, R.: Top-Down Versus Bottom-Up, Science, 328, 1241-1243, 2010.

Peischl, J., Ryerson, T. B., Brioude, J., Aikin, K. C., Andrews, A. E., Atlas, E., Blake, D., Daube, B. C., de Gouw, J. A., Dlugokencky, E., Frost, G. J., Gentner, D. R., Gilman, J. B., Goldstein, A. H., Harley, R. A., Holloway, J. S., Kofler, J., Kuster, W. C., Lang, P. M., Novelli, P. C., Santoni, G. W., Trainer, M., Wofsy, S. C., and Parrish, D. D.: Quantifying sources of methane using light alkanes in the Los Angeles basin, California, Journal of Geophysical Research-Atmospheres, 118, 4974-4990, 2013.

Peischl, J., Ryerson, T. B., Holloway, J. S., Parrish, D. D., Trainer, M., Frost, G. J., Aikin, K. C., Brown, S. S., Dube, W. P., Stark, H., and Fehsenfeld, F. C.: A top-down analysis of emissions from selected



Texas power plants during TexAQS 2000 and 2006, Journal of Geophysical Research-Atmospheres, 115, 15, 2010.

Petron, G., Karion, A., Sweeney, C., Miller, B. R., Montzka, S. A., Frost, G. J., Trainer, M., Tans, P., Andrews, A., Kofler, J., Helmig, D., Guenther, D., Dlugokencky, E., Lang, P., Newberger, T., Wolter, S., Hall, B., Novelli, P., Brewer, A., Conley, S., Hardesty, M., Banta, R., White, A., Noone, D., Wolfe, D., and Schnell, R.: A new look at methane and nonmethane hydrocarbon emissions from oil and natural gas operations in the Colorado Denver-Julesburg Basin, Journal of Geophysical Research-Atmospheres, 119, 6836-6852, 2014.

Quick, J. C.: Carbon dioxide emission tallies for 210 U.S. coal-fired power plants: A comparison of two accounting methods, J. Air Waste Manage. Assoc., 64, 73-79, 2014.

Raupach, M. R. and Legg, B. J.: THE USES AND LIMITATIONS OF FLUX-GRADIENT RELATIONSHIPS IN MICROMETEOROLOGY, Agric. Water Manage., 8, 119-131, 1984.

Ritter, J. A., Barrick, J. D. W., Watson, C. E., Sachse, G. W., Gregory, G. L., Anderson, B. E., Woerner, M. A., and Collins, J. E.: AIRBORNE BOUNDARY-LAYER FLUX MEASUREMENTS OF TRACE SPECIES OVER CANADIAN BOREAL FOREST AND NORTHERN WETLAND REGIONS, Journal of Geophysical Research-Atmospheres, 99, 1671-1685, 1994.

Roscioli, J. R., Yacovitch, T. I., Floerchinger, C., Mitchell, A. L., Tkacik, D. S., Subramanian, R., Martinez, D. M., Vaughn, T. L., Williams, L., Zimmerle, D., Robinson, A. L., Herndon, S. C., and Marchese, A. J.: Measurements of methane emissions from natural gas gathering facilities and processing plants: measurement methods, Atmos. Meas. Tech., 8, 2017-2035, 2015.

Ryerson, T. B., Buhr, M. P., Frost, G. J., Goldan, P. D., Holloway, J. S., Hubler, G., Jobson, B. T., Kuster, W. C., McKeen, S. A., Parrish, D. D., Roberts, J. M., Sueper, D. T., Trainer, M., Williams, J., and Fehsenfeld, F. C.: Emissions lifetimes and ozone formation in power plant plumes, Journal of Geophysical Research-Atmospheres, 103, 22569-22583, 1998.

Stull, R. B.: An Introduction to Boundary Layer Meteorology, Kluwer Academic Publishers, 1988.

Taylor, G. I.: Diffusion by Continuous Movements, Proceedings of the London Mathematical Society, s2-20, 196-212, 1922.

Tratt, D. M., Buckland, K. N., Hall, J. L., Johnson, P. D., Keim, E. R., Leifer, I., Westberg, K., and Young, S. J.: Airborne visualization and quantification of discrete methane sources in the environment, Remote Sensing of Environment, 154, 74-88, 2014.

Turnbull, J. C., Karion, A., Fischer, M. L., Faloona, I., Guilderson, T., Lehman, S. J., Miller, B. R., Miller, J. B., Montzka, S., Sherwood, T., Saripalli, S., Sweeney, C., and Tans, P. P.: Assessment of fossil fuel carbon dioxide and other anthropogenic trace gas emissions from airborne measurements over Sacramento, California in spring 2009, Atmospheric Chemistry and Physics, 11, 705-721, 2011.

Wecht, K. J., Jacob, D. J., Frankenberg, C., Jiang, Z., and Blake, D. R.: Mapping of North American methane emissions with high spatial resolution by inversion of SCIAMACHY satellite data, Journal of Geophysical Research-Atmospheres, 119, 7741-7756, 2014.



Weil, J. C.: Dispersion in the Convective Boundary Layer. In: Lectures on Air Pollution Modeling, Venkatram, A. and Wyngaard, J. C. (Eds.), American Meteorological Society, Boston, MA, 1988.

Weil, J. C., Sullivan, P. P., Patton, E. G., and Moeng, C.-H.: Statistical Variability of Dispersion in the Convective Boundary Layer: Ensembles of Simulations and Observations, Boundary-Layer Meteorology, 145, 185-210, 2012.

Willis, G. E. and Deardorff, J. W.: LABORATORY MODEL OF DIFFUSION INTO CONVECTIVE PLANETARY BOUNDARY-LAYER, Quarterly Journal of the Royal Meteorological Society, 102, 427-445, 1976.

Wilson, J. D. and Shum, W. K. N.: A REEXAMINATION OF THE INTEGRATED HORIZONTAL FLUX METHOD FOR ESTIMATING VOLATILIZATION FROM CIRCULAR PLOTS, Agric. For. Meteorol., 57, 281-295, 1992.

Wratt, D. S., Gimson, N. R., Brailsford, G. W., Lassey, K. R., Bromley, A. M., and Bell, M. J.: Estimating regional methane emissions from agriculture using aircraft measurements of concentration profiles, Atmospheric Environment, 35, 497-508, 2001.

Yacovitch, T. I., Herndon, S. C., Roscioli, J. R., Floerchinger, C., McGovern, R. M., Agnese, M., Petron, G., Kofler, J., Sweeney, C., Karion, A., Conley, S. A., Kort, E. A., Nahle, L., Fischer, M., Hildebrandt, L., Koeth, J., McManus, J. B., Nelson, D. D., Zahniser, M. S., and Kolb, C. E.: Demonstration of an Ethane Spectrometer for Methane Source Identification, Environmental Science & Technology, 48, 8028-8034, 2014.

Yuan, B., Kaser, L., Karl, T., Graus, M., Peischl, J., Campos, T. L., Shertz, S., Apel, E. C., Hornbrook, R. S., Hills, A., Gilman, J. B., Lerner, B. M., Warneke, C., Flocke, F. M., Ryerson, T. B., Guenther, A. B., and de Gouw, J. A.: Airborne flux measurements of methane and volatile organic compounds over the Haynesville and Marcellus shale gas production regions, Journal of Geophysical Research-Atmospheres, 120, 6271-6289, 2015.