# Peer review of "Application of Gauss's Theorem to quantify localized surface emissions from airborne measurements of wind and trace gases"

_Atmospheric Measurement Techniques, 2017_

## Referee Comment (RC1) · Anonymous Referee #1 · 24 May 2017

The paper by Conley et al. describes the use of aircraft-borne in-situ measurements for the quantification of localized greenhouse gas sources in a heterogenous field of potential sources. Overall, the paper is well written and the well described theoretical method may be a powerful tool to improve quantification of greenhouse gas emissions, especially in a complex area. My main point to criticize is that there is a missing link between the performed LES simulations and the presented aircraft measurements. Therefore it is not totally clear to me whether you use the LES simulations to show whether the suggested flight pattern is suitable in general, or if you use the simulations to actually design the flight pattern (e.g. the loop diameter) for each single mission. Independent from this, I´d suggest to focus on one specific flight/flight series throughout the paper. This would simply help to evaluate how good the LES simulations agree with the observations and e.g. how useful the simulations are to optimize the circling radius etc.

Also, a summarizing paragraph ("Summary/Conclusions/Outlook") is missing (by accident?) and essentially needs to be added.

I suggest publication after the following points have been addressed:

**Main comments:**

Line 141: According to table 1, you release the emissions in a box of 50x50mx8m (table 1). What means the question mark after 8 in the column dz? Especially the (center of the) release height is a very critical parameter. Have you done sensitivity studies by varying the release height to e.g. account for buoyancy? I assume the release rate is constant after the start of the release? Why is the release rate so small (~3kg/h), especially compared to the stated detection limits of 5 kg/h?

Figure 1/3: See my main point: Is there any possibility to combine/compare both figures in order to see how good the plume shape is represented in the model, compared to the (lower resolved) measurements? At least, you should be able to virtually fly along the flight path, extract the concentration levels and plot it together with the measurements along the time-series of the flight (although I know that this kind of graphic representation may be misleading if the plume is slightly shifted).

Line 299ff/Figure 6 and 7: See my main point: Why don´t you compare the LES results with the Aerodyne real test case? I´d suggest using the same flight example for both simulations and measurements (you may also discuss the variability based on your set of simulations).

**Minor comments:**

Line 111: I assume that the flow rate is controlled in a way that the lag time of both instruments is independent from the ambient pressure?

Figure 1: Please provide more details such as date/time/duration of flight, (derived) source strength, loop diameter.

Line 300: What means similar? The number of passes?

Line 374: Please give the uncertainty of the release rate.

**Technical comments:**

Please check the number of the equation in section 3.8

Figure 10: What is the unit of the x-axis? Please correct "geographic distribution of methane" to "ethane" in the caption. Please increase the dot size in the right figure.

---

## Referee Comment (RC2) · Anonymous Referee #3 · 20 Jun 2017

The paper by Conley et al. presents and validates a technique to infer point-source emission rates from in-situ aircraft observations of the atmospheric concentrations in a cylindrical volume around the source. The techniques is approached theoretically through LES modelling as well as experimentally through the analysis of actual aircraft observations. The observation part is, however, rather short deserving more detailled discussions.

The writing of the paper is somewhat unclear in some places and, it appears like the manuscript lacks some text at the end wrt. discussing Figure 8-10 and conclusions. I recommend reworking the manuscript for clarity along my comments below and extending the discussion of Figures 8-10.

Comments

line 103, throughout the manuscript: Check usage of "\citep" vs. "\citet".

line 137: Check usage of "ABL" vs. "CBL".

line 155: Several symbols undefined. Use real equations instead of in-line math.

line 176-178: I do not understand what the paragraph refers to. Isn't it redundant?

line 199: "the two terms that make up the path integral in equation (5)". There is only one term in equation (5), the horizontal divergence term dropped out before. In general, the ordering of equ. (1) through (6) appears confusing. Please check whether section 3.1 can be improved wrt. clarity.

line 200: Are all symbols defined? Use real equations instead of in-line math.

line 235: What ist U, what is T?

line 254: "Assuming." Remove.

line 258: Explain what a cospectrum is.

section 3.4: Rework for clarity considering background of the general reader of AMT.

line 331: This paragraph explains the mass derivative term in equation (6). Mass m, however, does not occur in any of the explanations. What is "beta"? Why is the time rate of change of mass an uncertainty per se that needs to be added in quadrature? Please rework this paragraph for clarity.

line 415: The version of the manuscript I reviewed (downloaded from the AMT website) ends with "rendering this technique completely unsuitable." I wonder whether this is actually intended to be a take-home message of the manuscript. Should there be a conclusion section which was accidentally missed out?

[Figure]

Table 1: What are the question marks?

Tables 2 and 3: Please use date formatting that is unambiguous for international readers (e.g. YYYY-MMM(string)-DD).

Table 2: hr-11 -> hr-1. The difference column refers to ethane.

Table 3: The difference column refers to methane, right? Please make this clear.

Table 4: Why is there the additional column "Hour"? What is the unit of the $CO_2$ emssion rate ("T"?)?

Figure 1: "to right" -> "to the right."

Figure 2: "Equation (4)" Isn't it equation (3)?

Figure 3: The title and color bar label of the figure "Cross wind concentration" are misleading since the contours represent the integral in cross wind direction not any kind of cross wind dimension.

Figure 4: Please explain in more detail what a cospectrum is.

Figure 5: Define "d" in mathematical terms and relate it to one of your equations. Is "Q" the same as "Q_c" in the text?

Figure 6: "This figure shows" . . . unnecessary to say in a figure caption.

Figure 8, 9, 10: Are these figures ever used/referred to in the manuscript? Please discuss in detail.

---

## Author Comment (AC1) · 20 Jul 2017

The comment was uploaded in the form of a supplement:
https://www.atmos-meas-tech-discuss.net/amt-2017-55/amt-2017-55-AC1-supplement.pdf

---

## Author Comment (AC3) · 20 Jul 2017

**Application of Gauss's Theorem to quantify localized surface emissions from airborne measurements of wind and trace gases**

Stephen Conley\*1,6, Ian Faloona1, Shobhit Mehrotra1, Maxime Suard1, Donald H. Lenschow2, Colm Sweeney4, Scott Herndon3, Stefan Schwietzke4,5, Gabrielle Pétron4,5, Justin Pifer6, Eric A. Kort7 and Russell Schnell5

1Department of Land, Air, & Water Resources, University of California, Davis, 95616, USA 2Mesoscale and Microscale Meteorology Laboratory, National Center for Atmospheric Research, Boulder, Colorado, 80307, USA

3Aerodyne Research, Inc, Billerica, Massachusetts, 01821, USA

[revised manuscript text omitted]
 | Date        | Laps | Released | Estimated       | Released | Estimated | Ethane     |
|------------|-------------|------|----------|-----------------|----------|-----------|------------|
| Location   |             |      | $CH_4$   | CH 4 | $C_2H_6$ | $C_2H_6$  | Difference |
|            |             |      | kg hr⁻¹  | kg hr⁻¹         | kg hr⁻¹  | kg hr⁻¹   |            |
| Colorado   | 2014-Nov-19 | 50   | 0.0      | -0.1±0.3        | 5.5±0.5  | 5.6±2.9   | +2%        |
| Arkansas   | 2015-Oct-03 | 19   | 0.0      | -3.4±12.3       | 8.1±0.8  | 10.0±6.1  | +24%       |

Table 2 - Ethane controlled releases.

| Experiment | Date        | Laps | Released | Estimated | Released | Estimated | Methane    |
|------------|-------------|------|----------|-----------|----------|-----------|------------|
| Location   |             |      | $CH_4$   | $CH_4$    | $C_2H_6$ | $C_2H_6$  | Difference |
|            |             |      | kg hr⁻¹  | kg hr⁻¹   | kg hr⁻¹  | kg hr⁻¹   |            |
| Rio Vista  | 2014-Nov-03 | 37   | 13.9±2.8 | 12.8±8.5  | 1.2±0.5  | 0.6±0.4   | -8%        |
| Rio Vista  | 2014-Nov-04 | 27   | 13.9±2.8 | 11.5±3.2  | 1.2±0.5  | 0.5±0.3   | -17%       |

Table 3 - Natural Gas controlled release

| Power Plant    | Date        | Hour
UTC | Laps | Reported $CO_2$
T hr -1 | Estimated $CO_2$
T hr -1 | Difference |
|----------------|-------------|-------------|------|---------------------------------------|----------------------------------------|------------|
| Rocky Mountain | 2014-Oct-06 | 20          | 19   | 99±14                                 | 111±24                                 | 13%        |
| Energy         |             |             |      |                                       |                                        |            |
| Saint Vrain    | 2014-Oct-04 | 19          | 21   | 124±17                                | 122±41                                 | -1%        |
| Pawnee         | 2014-Nov-19 | 20          | 14   | 575±81                                | 555±160                                | -3%        |
| Saint Vrain    | 2015-Sep-17 | 20          | 14   | 361±54                                | 280±115                                | -23%       |
| Four Corners   | 2015-Apr-11 | 18          | 12   | 1289±387                              | 1119±343                               | -13%       |
| Power Plant    |             |             |      |                                       |                                        |            |

Table 4 - Power Plant estimates. The mid-point of the measurements (hours UTC) is indicated in the third column(Hour). The reported emissions from the hour before to the hour after that time were averaged to derive the"Reported" emissions in column 5. Emissions are reported in units of metric tons (T) per hour.

**4. Acknowledgements**

Funding for this study provided by the California Energy Commission (CEC) and the US Department of Energy (DOE). Funding for the Denver and Arkansas portion of this work was provided by RPSEA/NETL contract no 12122-95/DE-AC26-07NT42677 to the Colorado School of Mines. Cost sharing was provided by Colorado Energy Research Collaboratory, the National Oceanic and Atmospheric Administration Climate Program Office, the National Science Foundation (CBET-1240584), Southwestern Energy, XTO, Chevron, Statoil and the American Gas Association, many of whom also provided operational data and/or site access. We also thank Professor Shuhua Chen for assistance with the WRF-LES code modifications and advice. The National Center for Atmospheric Research is sponsored by the National Science Foundation. This work was supported in part by the NOAA AC4 program under grant NA140AR0110139 and the Bureau of Land Management, Grant L15PG00058. We thank Ying Pan for her significant contribution to our understanding of the negative horizontal scalar flux.

---

## Author Response (AR1)

Response to Reviewer #1 of "Application of Gauss's Theorem to quantify localized surface emissions from airborne measurements of wind and trace gases" by Conley et al. submitted to Atmospheric Measurement Techniques 18-Apr-2017

General Comments

The paper by Conley et al. describes the use of aircraft-borne in-situ measurements for the quantification of localized greenhouse gas sources in a heterogenous field of potential sources. Overall, the paper is well written and the well described theoretical method may be a powerful tool to improve quantification of greenhouse gas emissions, especially in a complex area. My main point to criticize is that there is a missing link between the performed LES simulations and the presented aircraft measurements. Therefore it is not totally clear to me whether you use the LES simulations to show whether the suggested flight pattern is suitable in general, or if you use the simulations to actually design the flight pattern (e.g. the loop diameter) for each single mission. Independent from this, I´d suggest to focus on one specific flight/flight series throughout the paper. This would simply help to evaluate how good the LES simulations agree with the observations and e.g. how useful the simulations are to optimize the circling radius etc.

Also, a summarizing paragraph ("Summary/Conclusions/Outlook") is missing (by accident?) and essentially needs to be added.

We apologize for the unintentional obfuscation of the link between the LES computational results and the observational ones. The LES work was performed for just three different conditions as outlined in Table 1, and was not meant to specifically represent any of the individual observational surveys. Our intention is to very generally illustrate the dispersion of a plume in a convective boundary layer and to use those results to help guide the observational strategy development, such as optimal distance downwind and flux divergence profile extrapolation. We have added some text in the introduction (new line #103) to help clarify the approach:

*"Because the wind fields of turbulent flows cannot be predicted in detail, we do not attempt to compare specific features of our observations with specific LES results, but rather we use the numerical experiments to guide the development of the observational methodology. For example, by investigating the LES flux divergence profiles in the layer below the lowest flight altitude, we are able to estimate the contribution of this unmeasured component to the overall source strength."*

As recommended, we have added a Conclusions Section (Section 5) to the end of the manuscript to help summarize and point to further method development directions as recommended:

*This technique was developed out of the necessity to identify and quantify individual well pads in an extensive oil and gas production field. Consequently the frequent tracking of the upwind and downwind side of the source provides a very accurate determination of the location and magnitude of a given emission site. The main uncertainty arises from the effluent below the lowest flight altitude, but this is minimized by targeting a downwind distance determined by LES studies to provide very little change in the plume flux divergence from the lowest loop to the ground. In addition to the controlled release experiments, hundreds of sites have been measured using this technique with varying levels of success. Ideal conditions include flat terrain, ample sunlight to promote vertical mixing, consistent winds, and no nearby competing sources. Under optimal conditions we have demonstrated that measurement uncertainties are quite low, often better than 10%. As the conditions deteriorate from the ideal to situations involving complex terrain, variable winds or nearby upwind sources, measured uncertainties can increase to be as large or larger than the emission estimates themselves. In the worst case of stably stratified conditions (winter or night time), for instance, the lack of vertical mixing may preclude the trace gases emitted at the surface from reaching the minimum safe flight altitude. Complex terrain provides a challenge to the method because the aircraft is unable to maintain a constant altitude above the ground. A possible future refinement of this technique to be applied in complex terrain would be to fit the measurements of both wind and mixing ratio to a uniform 3-dimensional surface surrounding the source, where the grid passes through the terrain and then integrate the flux normal to this irregular virtual flight path. This would not assume level loop flight legs and would, in principle, account for individual loops being flown at differing altitudes and thus more closely track mass continuity near the terrain elevation.*

**Main comments:**

Line 141: According to table 1, you release the emissions in a box of 50x50mx8m (table 1). What means the question mark after 8 in the column dz?

Those were typos and have been removed.

Especially the (center of the) release height is a very critical parameter. Have you done sensitivity studies by varying the release height to e.g. account for buoyancy?

We have not experimented with elevated releases or lofting due to initial buoyancy of plumes. Because any elevated or buoyant release would only make the plume easier to detect from the aircraft, we considered a non-buoyant surface source as the limiting condition of detection. We feel that this complication is beyond the scope of this paper, but merits further investigation in the future.

I assume the release rate is constant after the start of the release? Why is the release rate so small (~3kg/h), especially compared to the stated detection limits of 5 kg/h?

Because LES is not subject to instrument noise or variability in the background - both of which determine the detection limit for actual measurements - the size of the release does not need

to be as large as in the actual atmosphere. The magnitude was chosen long before we had a very solid idea of what the actual measurement detection limit would eventually turn out to be.

Figure 1/3: See my main point: Is there any possibility to combine/compare both figures in order to see how good the plume shape is represented in the model, compared to the (lower resolved) measurements? At least, you should be able to virtually fly along the flight path, extract the concentration levels and plot it together with the measurements along the time-series of the flight (although I know that this kind of graphic representation may be misleading if the plume is slightly shifted). Line 299ff/Figure 6 and 7: See my main point: Why don´t you compare the LES results with the Aerodyne real test case? I´d suggest using the same flight example for both simulations and measurements (you may also discuss the variability based on your set of simulations).

As we have attempted to elaborate in the revised introduction (above), we do not feel that it is very informative to compare specific LES results with specific realizations observed in the real atmosphere. Because of the inherently random nature of turbulence these are bound to differ in their specific details (e.g., position or even structure of the effluent plume at any given instance.) While Figure 1 shows a sampled "snapshot" of the plume encountered during the ~30 minutes on station, the simulation results presented in Figure 3 are heavily averaged cross-wind integrated concentrations that are meant to illustrate the average structure of the plume in the downwind direction. Any single aircraft crossing of the plume is going to deviate from this picture, and therefore we feel that a direct comparison would not convey anything new or instructive.

In the case of Figure 6 and 7, again, we do not feel that a specific comparison between the details of the LES and observations is going to show anything informative. However, comparison of the Figures does show that the average behavior of the asymptotic approach to the actual emission rate appears similar.

**Minor comments:**

Line 111: I assume that the flow rate is controlled in a way that the lag time of both instruments is independent from the ambient pressure?

We have added the following text in Section 2.1 to clarify this point, "*Both lag times are slightly dependent on pressure, i.e., with a typical altitude change of ~1 km, the change in lag time is less than 10%, and is inconsequential when applying this method within a few hundred meters from the surface.*"

Figure 1: Please provide more details such as date/time/duration of flight, (derived) source strength, loop diameter.

We have added specific flight information to the caption as requested.  The revised caption of Figure 1 reads:

*Figure 1 - Map of the airplane flight pattern sampling a methane plume emanating from an underground storage facility.  Wind direction is indicated by the white arrow and the methane mixing ratio is given by the color bar to the right.  This flight was conducted on June 28, 2016 and took place between 12:46PM and 1:52PM LT at altitudes ranging from 91 m to 560 m with a loop diameter of approximately 3 km. The measured methane emission rate was 763±127 kg hr$^{-1}$.*

Line 300: What means similar? The number of passes?

We have deleted the word 'similar' to avoid confusion.

Line 374: Please give the uncertainty of the release rate.

An estimate of the uncertainty in the release rate has been added.

**Technical Comments:**

Please check the number of the equation in section 3.8

The equation in Section 3.8 has been eliminated in response to a recommendation by another reviewer.

Figure 10: What is the unit of the x-axis? Please correct "geographic distribution of methane" to "ethane" in the caption. Please increase the dot size in the right figure.

Figure 10 has become Figure 9 and the caption and axis labels have been expanded to clarify.

Response to Reviewer #3 of "Application of Gauss's Theorem to quantify localized surface emissions from airborne measurements of wind and trace gases" by Conley et al. submitted to Atmospheric Measurement Techniques 18-Apr-2017

The paper by Conley et al. presents and validates a technique to infer point-source emission rates from in-situ aircraft observations of the atmospheric concentrations in a cylindrical volume around the source. The techniques is approached theoretically through LES modelling as well as experimentally through the analysis of actual aircraft observations. The observation part is, however, rather short deserving more detailled discussions.

The writing of the paper is somewhat unclear in some places and, it appears like the manuscript lacks some text at the end wrt. discussing Figure 8-10 and conclusions. I recommend reworking the manuscript for clarity along my comments below and extending the discussion of Figures 8-10.

We thank the reviewer for the constructive comments and have attempted to clarify the discussion and expand the discussion of the last 3 figures in accordance with the recommendations below.

Comments

line 103, throughout the manuscript: Check usage of "\citep" vs. "\citet".

Done

line 137: Check usage of "ABL" vs. "CBL".

Done

line 155: Several symbols undefined. Use real equations instead of in-line math.

We removed the density in-line equation

line 176-178: I do not understand what the paragraph refers to. Isn't it redundant?

We have reordered and abridged the statements here to precede equation 3 in the hopes of making the mathematical procedures more clear.

line 199: "the two terms that make up the path integral in equation (5)". There is only one term in equation (5), the horizontal divergence term dropped out before. In general, the ordering of equ. (1) through (6) appears confusing. Please check whether section 3.1 can be improved wrt. clarity.

We apologize for the error, this was supposed to read equation 2 (not equation 5). We have rewritten Section 3.1 several times before submission and believe that this presentation is the most straight forward. The progression of equations 1-5 runs from the general governing equation (1), to picking apart the specific measurement terms in equations (2) & (3) and describing the approximations made in our analysis, and then reintroducing them into the governing equation in equations (4) & (5). We believe that the rewording of the discussion around equation (3), as per reviewer's suggestion, and the correction of the typo in line 199 helps eliminate unnecessary confusion.

line 200: Are all symbols defined? Use real equations instead of in-line math.

We removed the definition of the convective velocity scale, assuming that was common enough for a reader to know or easily look-up.

line 235: What ist U, what is T?

In rewriting Section 3.4 for clarity's sake, we removed the equation for mean advective heat flux, UT.

line 254: "Assuming." Remove.

Done.

line 258: Explain what a cospectrum is.

In discussing Figure 4 we included this description of the cospectrum, *"Because the integral of the cospectrum yields the total flux (covariance), this function is useful for examining the contributions to the total flux from each of the scales of motion (represented by aircraft speed divided by frequency)."*

section 3.4: Rework for clarity considering background of the general reader of AMT.

We have reworked this section on counter-directed turbulent fluxes to explain their origin and provide additional evidence for their existence from previous studies. We have included three new references (one being a textbook) surrounding the discussion of equation (8), and a reader who is not familiar with this type of turbulence reasoning can seek out these references to better understand the discussion.

line 331: This paragraph explains the mass derivative term in equation (6). Mass m, however, does not occur in any of the explanations. What is "beta"? Why is the time rate of change of mass an uncertainty per se that needs to be added in quadrature? Please rework this paragraph for clarity.

We have rewritten this section to improve clarity and removed the extraneous equation describing a linear regression fit.

line 415: The version of the manuscript I reviewed (downloaded from the AMT website) ends with "rendering this technique completely unsuitable." I wonder whether this is actually intended to be a take-home message of the manuscript. Should there be a conclusion section which was accidentally missed out?

We thank the reviewer for pointing out the awkward ending of the submitted discussion paper. We have added a short conclusions section to more clearly describe the limits of our method and potential avenues for future improvements.

Table 1: What are the question marks?

They were typos, we removed the question marks.

Tables 2 and 3: Please use date formatting that is unambiguous for international read- ers (e.g. YYYY-MMM(string)-DD).

Made the date labels unambiguous.

Table 2: hr-11 -> hr-1. The difference column refers to ethane.

Headers fixed.

Table 3: The difference column refers to methane, right? Please make this clear.

Difference header clarified.

Table 4: Why is there the additional column "Hour"? What is the unit of the $CO_2$ emssion rate ("T"?)?

Added caption text to define T = metric tons.

Figure 1: "to right" -> "to the right."

Done.

Figure 2: "Equation (4)" Isn't it equation (3)?

Corrected the equation referenced in caption.

Figure 3: The title and color bar label of the figure "Cross wind concentration" are misleading since the contours represent the integral in cross wind direction not any kind of cross wind dimension.

We have removed the colorbar labels and title.

Figure 4: Please explain in more detail what a cospectrum is.

In discussing Figure 4 we included this description of the cospectrum, *"Because the integral of the cospectrum yields the total flux (covariance), this function is useful for examining the contributions to the total flux from each of the scales of motion (represented by aircraft speed divided by frequency)."*

Figure 5: Define "d" in mathematical terms and relate it to one of your equations. Is "Q" the same as "Q_c" in the text?

We have described the variables more clearly in the caption.

Figure 6: "This figure shows" . . . unnecessary to say in a figure caption.

Removed redundant wording.

Figure 8, 9, 10: Are these figures ever used/referred to in the manuscript? Please discuss in detail.

We added text in the body of the manuscript to describe the results represented in these figures, and rearranged them slightly.